# Modeling water column gas transformation, migration and atmospheric flux from seafloor seepage

Knut Ola Dølven[1,2], Håvard Espenes[3,4,*], Alfred Hanssen[1,*], Muhammed Fatih Sert[1], Magnus Drivdal[3], Achim Randelhoff[3], and Bénédicte Ferré[1]

[1]Department of Geosciences, UiT The Arctic University of Norway, Tromsø, Norway
[2]Department of Electrical Engineering, UiT The Arctic University of Norway, Narvik, Norway
[3]Ocenaography section, Akvaplan NIVA, Tromsø, Norway
[4]SINTEF Ocean, Trondheim, Norway
[*]These authors contributed equally to this work.

**Correspondence:** Knut Ola Dølven (knut.o.dolven@uit.no)

**Abstract.** Understanding the fate of gas seeping from the seafloor is crucial for assessing the environmental impacts of both natural and anthropogenic seep systems, such as $CH_4$ cold seeps, leaking gas wells, and future carbon capture projects. We present a comprehensive modeling framework that integrates physical, chemical, and biological processes to estimate the 3-dimensional water column dissolved gas concentration field and 2-dimensional atmospheric flux field resulting from seafloor seeps. The framework consists of two main components: 1) a gas-phase model that calculates free gas dissolution and direct atmospheric release at the seep site, and 2) a concentration model that combines particle dispersion modeling with an adaptive-bandwidth kernel density estimator and customizable process modules. Applying the framework to a natural $CH_4$ seep at 200 m depth offshore northwestern Norway (May 20 - June 20, 2018), we found that dissolved methane was advected northeastward along the coast, spreading across shelves, reefs, and into fjord systems. Within days, the vertical $CH_4$ concentration profile was near inverted, with near-surface maxima, facilitating atmospheric exchange. Diffusive emissions covered large areas ($>10^5$ km$^2$) and was almost 3 times the local free gas flux. Around 0.7% of dissolved $CH_4$ reached the atmosphere during a 4 week period, microbial oxidation removed around 65%, while ~34% remained in the water column. Uncertainties caused by a range of model framework elements remain substantial, e.g. can estimates of microbial oxidation removal change from 65% to as low as 5.5% or as high as 91.4% depending on rate coefficient assumptions. Our framework provides a globally applicable tool that integrates free and dissolved gas dynamics and accommodates advanced hydrodynamic modeling. Its ability to explicitly resolve spatiotemporal fields enables the inclusion of complex physical and biogeochemical process modules and supports not only the quantification of atmospheric fluxes but also applications that require explicit field representations, such as assessing impacts on local ecosystems.

## 1 Introduction

Estimates of the contribution of seafloor gas seepage to atmospheric emissions and its impact on ocean environments are highly uncertain due to limited data and understanding of gas transformation and transport mechanisms in the water column.

Estimation of total atmospheric gas emissions from seep areas (e.g. Myhre et al., 2016) rely largely on either ship measurements or large-scale atmospheric inversion models. The former of these approaches only gives information on the local flux and requires some sort of up-scaling, while the latter is unable to estimate dispersed sources and/or weaker point sources precisely due to its rough scale and inability to completely decouple atmospheric sources from sinks (Thompson and Stohl, 2014). Quantifying dissolved gas in the water column usually involves measuring dissolved gas via water samples (e.g. Silyakova et al., 2020) or using *in situ* sensors (e.g. Gentz et al., 2014) which can be time-consuming and often result in poor data coverage. New modeling tools for constraining the environmental impacts of current and future seabed gas seepage from both natural and man made sources are therefore needed.

Gas released at the seabed can enter the atmosphere directly as free gas (bubbles) or via diffusive equilibrium of dissolved gas that has reached the sea surface. To estimate the total atmospheric emissions from a seabed seep and its dissolved distribution in the water column, one must be able to model both pathways simultaneously. Gas content in bubbles is constantly changing due to dissolution (gases in the bubble dissolve in the liquid) and exsolution (gases already dissolved in the liquid enter the bubble) driven by partial pressure gradients across the bubble rim. Additionally, chemical and biological processes can modify local dissolved gas content. Estimating the gas distribution in the water column and total atmospheric flux therefore requires a flexible framework which can integrate processes governing the gas phase dynamics and the hydrodynamics, accommodate atmospheric exchange, and other phenomena that modify water column gas content. Previous modelling efforts have typically focused on single gas phase frameworks including only selected processes (e.g McGinnis et al., 2006; Graves et al., 2015; Silyakova et al., 2020), however, key steps towards modeling the complete system have been made recently in Dissanayake et al. (2023) and Nordam et al. (2025). We aim to further expand on these studies from a methodological perspective and provide a pilot framework which can integrate all key processes governing free and dissolved transport and transformation of seeped gas and give a realistic estimate of the time varying 3-dimensional (3D) water column concentration field and 2-dimensional (2D) atmospheric release field.

Our approach integrates a gas phase model with a hydrodynamic model using particle dispersion modeling (similar to Dissanayake et al., 2023). It estimates the 3D distribution of gas in the water column and the total (free and diffusive) atmospheric 2D gas release resulting from observed seabed seepage. This approach offer flexible inclusion of atmospheric flux and chemical and biological process modules affecting dissolved gas content in the water column. Explicit concentrations (molecules per volume) are obtained using kernel density estimation. Atmospheric dissolved flux estimates are obtained using a bulk model and atmospheric free gas flux via a gas phase model. We tested the framework by quantifying direct and diffusive atmospheric fluxes as well as 3D dissolved gas distribution between May 20 and June 20, 2018 for a methane ($CH_4$) seep area offshore Northwestern Norway.

## 2 Method

Our goals are two-fold: i) Calculate the combined total amount of seep-derived gas that reaches the atmosphere - both direct free gas release and ventilation of dissolved gas, and ii) Estimate the impact of seeped gas on the scalar dissolved gas concentration

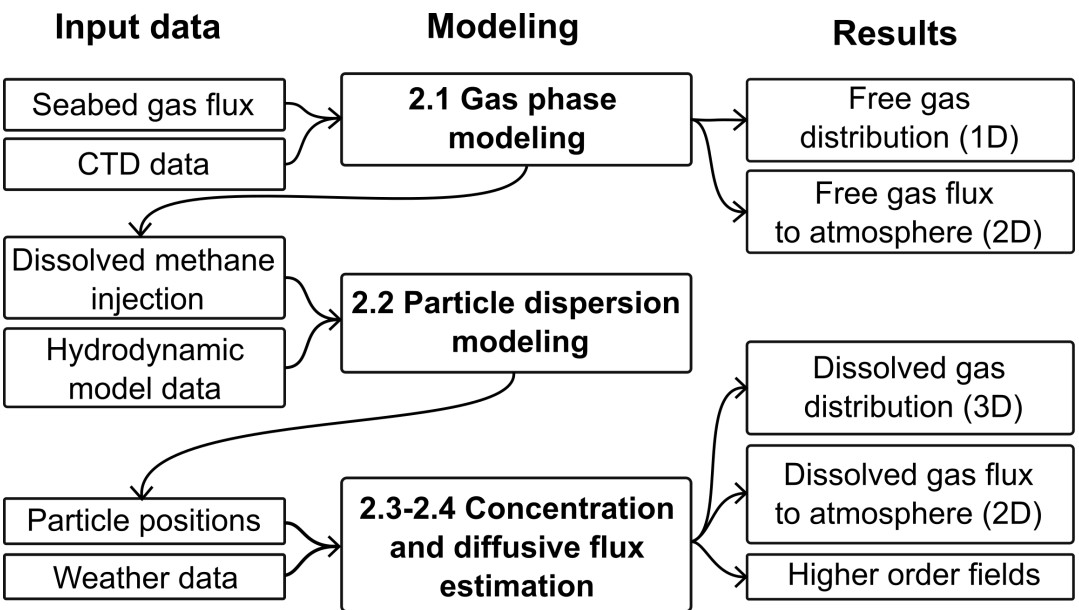

**Figure 1.** Model framework flowchart. The emboldened modeling steps and associated numbers refer to the four subsections of the Methods section. The "D" in the results column refers to spatial dimensions.

field, i.e., we seek the anomaly $\Phi'(x,y,z,t)=\Phi(x,y,z,t) - \Phi_0(x,y,z,t)$ caused by the seeps, where $\Phi(x,y,z,t)$ denotes the total concentration and $\Phi_0(x,y,z,t)$ is a background concentration.

The outlined goals are achieved by adapting and integrating existing and new models, of which output from one model serves as a final result or feeds another model. We first use seabed gas volume flux data and a two-phase gas model to calculate the gas dissolution rates and direct atmospheric gas (bubble) release. Dissolved injection rate output from the gas phase model then feeds a concentration model that combines an existing dispersion modeling framework with an adaptive kernel density estimator, including an atmospheric flux module and options for water column process modules. Figure 1 shows the complete framework, with input data in the left column, the modeling steps in the center column, and the final results in the right column. Each modeling section is detailed in the corresponding subsection.

## 2.1 Gas phase modeling

Free atmospheric gas fluxes and dissolved gas profiles injected to the water column are initially modeled for each observed seep using the seabed free gas flux data and the M2PG1 gas phase model (Jansson et al., 2019). M2PG1 provides an integrated solution of dissolved and free gas in a 1-dimensional water column, with sources and sinks at both horizontal and vertical model boundaries. It simultaneously models gas exchange, dissolution, and associated dissolved gas concentration of five gas species (methane ($CH_4$), Argon (Ar), Carbon dioxide ($CO_2$), Nitrogen ($N_2$), and Oxygen ($O_2$)) across a user-defined initial spectrum of bubble sizes. The bubble size spectrum and gas distribution across this spectrum vary freely across the spatio-

temporal model domain. The model includes several bubble shape and rising speed models, microbial oxidation of $CH_4$ using first order kinetics (Griffiths et al., 1982; Chan et al., 2019), diffusive exchange with the atmosphere, dissolved transport due to vertical turbulent exchange of water masses, as well as loss due to advection across the model boundary (Jansson et al., 2019). While dynamic solutions are permissible in M2PG1, we have opted for a steady state solution in our modeling framework.

The input parameters include seabed gas flux, bubble characteristics (size distribution, rising speed, dirtiness, flatness), temperature, salinity, microbial $CH_4$ oxidation rate coefficients (MOx), ambient dissolved gas concentrations (for all five gases), vertical mixing (turbulent) and local ocean currents. Seabed free gas flux data can in theory be obtained by any means available, although hydroacoustics have been used extensively due to its relatively straightforward deployability and large coverage (e.g. Ferré et al., 2020).

In our implementation of M2PG1 we used a new estimation technique to determine the horizontal model domain size in M2PG1. Horizontal domain size was previously chosen ambiguously in M2PG1 (Jansson et al., 2019) and could cause significant exchange rate errors. Our method removes this ambiguity by estimating the horizontal bubble plume extent based on local conditions. Details are provided in Appendix A.

The steady-state output from the M2PG1 simulation provides two key results: i) direct atmospheric gas flux and ii) injection

rates of dissolved gas to the surrounding water column. The former is a direct output in M2PG1 and the latter, which are key input for the concentration modeling steps (Sections 2.3 – 2-4), can be derived from the dissolved gas concentration profiles by calculating the dissolved gas loss $q$ [mol s$^{-1}$] to the water column at the downstream boundary of each M2PG1 grid cell. The steady-state mass flux assumption gives:

$$q = A_{\mathcal{M}}^{\perp} U(\varphi_{\mathcal{M}} - \varphi_b), \tag{1}$$

where $A_{\mathcal{M}}^{\perp}$ [m$^2$] is the vertical grid cell area (Appendix A), $U$[m s$^{-1}$] the current speed, $\varphi_{\mathcal{M}}$ [mol m$^{-3}$] the estimated concentration within the grid cell and $\varphi_b$ [mol m$^{-3}$] the assumed concentration at the upstream boundary.

## 2.2   Particle dispersion modeling

To estimate the unobservable dissolved gas concentration field anomaly $\Phi'(x, y, z, t)$, we must model the advection and spread of the dissolved gas from the seeps. We chose to simulate the transport and dispersion of the gas from the release site using

OpenDrift, which is a Lagrangian particle trajectory modeling software (Dagestad et al., 2018). In practice, this means that we distribute (virtually) the released $CH_4$ over a discrete number of virtual particles, and update the particle positions at discrete times $t_n$ for time-steps $n = 1, 2, 3, ..., N$ according to the output from a hydrodynamic model. Each timestep is separated by a time interval $\Delta t$. We then define $S[n]$ as the number of virtual particles seeded at the modeled seep sites at each time step. This generates a total of $Z = \sum_{n=1}^{N} S[n]$ particles indexed by $\zeta = 1, 2, 3, ..., Z$. Note that we throughout this manuscript will use

square "[·]" versus round "(·)" brackets to distinguish between discrete and continuous spatiotemporal arguments, respectively.

Once particles are seeded, OpenDrift calculates the trajectory of each particle individually by numerically solving a stochastic differential equation which is consistent with the Lagrangian representation of the advection-diffusion equation (see e.g.

Spivakovskaya et al. (2007)). The drift in particle position $\boldsymbol{\eta}$ can be expressed as

$$d\boldsymbol{\eta} = \boldsymbol{U}_\mu(\boldsymbol{\eta},t)dt + \boldsymbol{B}(\boldsymbol{\eta},t)d\boldsymbol{W}(t) \tag{2}$$

where $\mathbf{U}_\mu(\boldsymbol{\eta},t)$ represents displacement produced by the underlying (mean) velocity field and the second term represents displacement from random, diffusive processes and is composed of a diffusivity matrix $\boldsymbol{B}(\boldsymbol{\eta},t)$ and increments of a Wiener process $d\boldsymbol{W}(t)$. The advective term ($\mathbf{U}_\mu(\boldsymbol{\eta},t)$) is determined by velocity fields obtained from the hydrodynamic model. OpenDrift represents the diffusivity $\boldsymbol{B}(\boldsymbol{\eta},t)$ as a diagonal matrix with a horizontal and a vertical diffusivity. If available, these diffusivities can be directly read from the hydrodynamic model output. Otherwise, OpenDrift can also estimate the diffusivity

coefficients using one of several built-in parametrizations. Finally, OpenDrift returns individual (traceable) positions $\boldsymbol{\eta}_\zeta[n]$ for each seeded particle at each time-step they spend in the model domain.

### 2.2.1 Particle mass

To associate the particle distribution with dissolved gas content, we latch a *particle mass* $\Gamma_\zeta$ to each seeded particle, which explicitly corresponds to the number of moles each particle represents (this mass has no influence on the particle buoyancy).

Each particle is thus interpreted as a virtual single-point representation of some local spatial distribution of $\Gamma_\zeta$ moles of dissolved gas molecules.

 The initial mass, i.e. mass at release, of an arbitrary particle $\zeta$ is scaled such that the total released particle mass from all modeled seeps combined at timestep $n$ approximates the total number of moles of gas dissolved in the water column during the time interval $\Delta t$ centered on $t_n$. In practice, we distribute the integrated sum of modeled (using the gas phase model) injected

gas molecules from $t_n - \Delta t/2$ to $t_n + \Delta t/2$ evenly over the seeded particles. The mass of particle $\Gamma_\zeta$ seeded at time-step $n$ is then obtained by

$$\Gamma_\zeta[n] = \frac{\Delta t \sum_{p=0}^{P} \Upsilon_p[n]}{S[n]} \tag{3}$$

where $\Upsilon_1[n], \Upsilon_2[n], \ldots, \Upsilon_P[n]$ [mol s$^{-1}$] are total injected dissolved gas from all $P$ modeled seeps. Approximation to the modeled dissolved gas release profiles at each modeled seep is achieved by seeding different amount of particles at different

depths. Particle masses are then subsequently individually adjusted at each time-step to simulate processes affecting gas content. Each particle thus has a successively constructed mass time-series, where the current mass $\Gamma_\zeta[n]$ is determined by the previous mass $\Gamma_\zeta[n-1]$ and selected mass modification functions.

### 2.2.2 Particle count

Our framework must be able to model an extensive 3-dimensional domain (e.g. larger ocean regions), making computational

complexity a challenge. Both computation time and estimation quality increase with the number of active particles present in the domain. This makes it crucial to be able to strike a decent compromise between the two, which typically involves removal of particles that have been present in the domain for a certain number of time-steps. Total particle count $\Lambda[n]$ in the domain

can be expressed as

$$\Lambda[n] = \Lambda[n-1] + S[n] - L[n] - \wp[n], \tag{4}$$

where $L[n]$ is the number of particles leaving the modelled geographical domain, and $\wp[n]$ the number of removed particles. A constant particle count is obtained when $S[n] \sim L[n] + \wp[n]$. Since $\wp[n]$ represents non-physical loss of gas, the model simulation would ideally run with a spin-up time that ensures $S[n] \sim L[n]$. Unfortunately, this typically results in unreasonable computation times and/or spin-up periods, making particle removal necessary. To limit errors caused by removed particles, we apply a function that redistributes mass from all removed particles to nearby non-removed particles. The redistribution is weighted according to the inverse distance from the removed particle within a user defined distance limit $d_{max}$, giving a non-removed particle $\tau$ an added mass of

$$\gamma_\tau = \Gamma_\theta \frac{||\boldsymbol{\eta}_\tau - \boldsymbol{\eta}_\theta||_2^{-1}}{\sum_{\zeta \in \mathcal{T}} ||\boldsymbol{\eta}_\zeta - \boldsymbol{\eta}_\theta||_2^{-1}} \tag{5}$$

from the removed particle $\theta$. Here, $\tau \in \mathcal{T}$ and $\mathcal{T}$ is the set of non-removed particles with indices $\zeta$ satisfying $||\boldsymbol{\eta}_\zeta - \boldsymbol{\eta}_\theta||_2 \leq d_{max}$, and $||\cdot||_2$ denotes the Euclidean norm. This solution changes the problem of non-physical loss of dissolved gas to one of non-physical redistribution. This can affect model results by shifting particle mass towards the seed location, since the density of particles are in general higher closer to the release point. However, we consider this artifact less problematic than mass simply disappearing.

## 2.3 Grid Projected Adaptive-bandwidth Kernel Density Estimator

Having an explicit relationship between dissolved gas content (of seep origin) and particle mass $\Gamma_\zeta$ allows us to infer gas concentrations by evaluating the particle mass per unit volume, which we refer to as the particle density. Let us assume that the particles $\zeta = 0, 1, 2, ..., Z$ are scaled/weighted samples (using their mass $\Gamma_\zeta$) from an unknown, smooth, underlying particle density field $\phi(x, y, z, t)$ which approximates the seep-induced gas concentration anomaly field $\Phi'(x, y, z, t)$. Estimation of $\Phi'(x, y, z, t)$ can then be done via the estimate $\widehat{\phi}$ of $\phi(x, y, z, t)$ using the particle data set.

To get $\widehat{\phi}$, we employ a discrete spatiotemporal grid $[i, j, k, n]$, where $i = 1, 2, .., I$, $j = 1, 2, .., J$, $k = 1, 2, ..K$, $n = 0, 1, 2, .., N$, and $I, J, K, N$ denote the number of grid cells in east, north, vertical and temporal dimensions, respectively. Grid cell center positions are given by $[x_i, y_j, z_k, t_n]$, with horizontal resolution $\Delta\lambda$ (in both directions), vertical resolution $\Delta z$, and temporal resolution $\Delta t$. We then bin all mass in the temporal and vertical domains and obtain separate estimates $\widehat{\phi}[i, j]$ of $\phi(x_i, y_j)$ for each resulting depth layer $k$ and time-step $n$ to form the final estimate $\widehat{\phi}[i, j, k, n]$. Obtaining $\widehat{\phi}[i, j, k, n]$ thus translates to solving a series of 2-dimensional density estimation problems (see e.g. Silverman, 1986).

Due to the extensive model domain and the need to obtain one estimate for every depth layer and time-step ($K \times N$ estimates), our density estimator needs to be fast and allow reliable density estimates from limited particle counts. It must also handle regions with low and high concentrations and concentration gradients as well as complex boundaries like fjords and islands. A commonly used density estimator in similar contexts is the histogram estimator, which unfortunately has several well-known limitations in these applications (the histogram estimator and its drawbacks are detailed in Appendix B). Previous studies on

concentration estimation from particle dispersion model data have shown that Kernel Density Estimators (KDEs) can offer far superior information exploitation than the histogram estimator and overcome many of its drawbacks (see e.g De Haan, 1999; Vitali et al., 2006; Björnham et al., 2015; Barbero et al., 2024; Yang et al., 2026). One remaining challenge in our specific application, however, is the lack of available KDEs tailored to coastal ocean regions that appropriately adapt to spatial density variability (adaptive bandwidth) and complex boundary geometries (bathymetry). We therefore formulated a new 2-dimensional adaptive-bandwidth KDE to provide our density estimates.

Kernel density estimation is a standard non-parametric way to estimate the density of a random variable using kernel functions (Silverman, 1986). This offers density estimates that are differentiable, grid cell size independent and generally more realistic than histogram estimators, without lower density limitations. In our case, a kernel density estimate involves placing a symmetric, smooth, and weighted kernel function at each particle position. By summing up the kernel contributions, the density field $\widehat{\phi}$ at position $\boldsymbol{r}_0$ located within the volume $V$ can be estimated via the general kernel density estimator formula

$$\widehat{\phi}(\boldsymbol{r}_0) = \frac{1}{V}\sum_{\zeta=1}^{Z}\Gamma_\zeta K_h\left(\|\boldsymbol{r}_0 - \boldsymbol{\eta}_\zeta\|_2\right). \tag{6}$$

Here, $K_h(\xi) \equiv (1/h)K(\xi/h)$, where $K(\xi)$ is a non-negative, normalized, and symmetric kernel function, $h$ a bandwidth (smoothing) parameter, and $\boldsymbol{r}_0$ the estimate position.

It is well established that the choice of kernel shape $K(\xi)$ is of less importance, as long as it adheres to the kernel function requirements. We define the base kernel $K(\xi)$ as a standardized 2-dimensional Gaussian, i.e. $K(\xi) = \exp\left(-\xi^2/2\right)/2\pi$.

Selecting an appropriate bandwidth $h$ is crucial, as a poor choice can cause large errors (De Haan, 1999), particularly due to over-smoothing (Larsen et al., 2002). Several methods exist for selecting $h$ by evaluating the statistical properties of the collected data, but they typically rely on strict assumptions on the underlying field. For heterogeneous fields, such as ours, where statistical properties vary across the domain, local adaptation of $h$ is necessary to give realistic density estimates in both high and low particle count areas. Furthermore, the presence of complex boundaries in the form of bathymetry and coastlines introduces additional challenges, both for providing valid estimates and for computational complexity. To handle these challenges, we have proposed a KDE that is bathymetry bounded and estimates a locally adapted kernel bandwidth $h$ using an expanded version of Silverman's rule (Silverman, 1986) which accommodates correlated, weighted data. Computational complexity is constrained via grid-projection and pre-computation of kernels. Testing and validation of the estimator were done using synthetic simulations (see Appendix C).

### 2.3.1 Grid projection and pre-computed kernels

To improve computational times, we have implemented a grid-projected estimator (Sole-Mari et al., 2019). This involves obtaining a preliminary density $\tilde{\phi}[i,j]$ using the histogram estimator via Eq. (B1), i.e. calculate the accumulated particle mass of all particles within each grid cell. All mass then belongs to a discrete grid, where any difference in position $\mathbf{r}$ between two locations of interest $[i,j]$ and $[i_0, j_0]$ can be expressed as $(i - i_0, j - j_0)\Delta\lambda$.

Furthermore, we pre-compute a set of normalized kernels with fixed, discrete bandwidths given by:

$$h[\omega] = \frac{\omega \Delta \lambda}{3}, \quad \text{where} \quad \omega = 0, 1, 2, ... \Omega. \tag{7}$$

Each initial non-discrete bandwidth estimate $h'$ from the data-driven bandwidth algorithm is then mapped to the nearest candidate in $h[\omega]$. Kernel support is set to $\pm \omega \Delta \lambda$ beyond which the kernel contribution is set to zero. Leaked kernel mass is added back across the kernel domain using the Kernel function. These simplifications make complexity scale with particle-containing cells instead of particles. It also allows for fast vectorized operations which drastically reduce computation time while giving negligible errors for large grids (Sole-Mari et al., 2019).

### 2.3.2 Data-driven adaptive bandwidth selector

The conventional Silverman's rule of thumb (Silverman, 1986) selects the optimal bandwidth $h$ that minimizes the integrated mean square error under the assumption of Gaussian distributed data with variance $\sigma^2$. For a $d$-dimensional Gaussian kernel, one obtains

$$h = \left( \frac{4}{d+2} \right)^{\frac{1}{d+4}} N^{-\frac{1}{d+4}} \sigma, \tag{8}$$

where $N$ is the number of data samples, i.e., the number of particles. For $d = 2$, the expression simplifies to

$$h = N^{-\frac{1}{6}} \sigma. \tag{9}$$

Here we modify Silverman's rule to yield reasonable estimates for our multi-modal, correlated, non-homogeneous, weighted data set by: i) adapting $h$ locally for a square shaped horizontal "adaptation" grid of size $P \times P$ surrounding each particle containing grid cell where we assume near normal, unimodal distribution, ii) estimating the *effective* (uncorrelated) sample size $N_{\text{eff}}$ and iii) implementing bias corrections to a $\sigma$-estimator for weighted data-sets. The size $P$ is determined by an integral length scale estimate (as outlined below) of the entire $I \times J$ 2-d grid. For an arbitrary cell $[i, j]$, the adaptation grid is defined by letting $l$ and $m$ be discrete indices on the grid such that $1 \le l, m \le P$ with step size $\Delta \lambda$ in both directions. The number of particles contained in the local grid (prior to grid projection) is denoted as $N_g$, and the pre-computed histogram density $\tilde{\phi}$ serves as the underlying data for obtaining estimates of the local $h$. We will now describe the procedure of obtaining estimates of $N_{\text{eff}}$ and $\sigma$ to get the local $h$ for an arbitrary adaptation grid.

Spatial correlations present in environmental data decrease the effective degrees of freedom in the sample set and we must therefore estimate and use the effective sample size $N_{\text{eff}}$ to obtain a reasonable $h$ for the local grid (e.g. Larsen et al., 2002). To estimate the effective number of spatially uncorrelated samples, a measure of correlation length is employed. We use the so-called *integral length scale* known from turbulence theory and statistical physics, see e.g., (Yaglom, 1987; Frisch, 1995; Pécseli, 2000) as our objective measure of the correlation length. The correlation length $L_c$ in terms of the integral length scale is formally defined by Frisch (1995)

$$L_c = \frac{\int_0^\infty |R(\varrho)| d\varrho}{R(0)}, \tag{10}$$

where $R(\varrho) = \mathbb{E}\{\phi(x)\phi(x+\varrho)\}$ is the spatial autocorrelation function (ACF) of a continuous univariate spatial random process $\phi(x)$, $\varrho$ is a spatial lag coordinate, and $\mathbb{E}\{\cdot\}$ is the statistical expectation operator.

We now proceed by defining a *local integral length* scale $L_c$ for the binned adaptation window. First, we estimate the local one-dimensional ACF of $\tilde{\phi}$ along each row $l$ and column $m$ of the square grid by using the standard unbiased ACF estimator (e.g., Percival & Walden, 1993)

$$\widehat{R}_l^{row}[\lambda] = \frac{1}{P-\lambda}\sum_{m=1}^{P-\lambda} \tilde{\phi}_{l,m}\tilde{\phi}_{l,m+\lambda} \qquad \text{for} \quad l = 1,2,...,P \tag{11}$$

$$\widehat{R}_m^{col}[\lambda] = \frac{1}{P-\lambda}\sum_{l=1}^{P-\lambda} \tilde{\phi}_{l,m}\tilde{\phi}_{l+\lambda,m} \qquad \text{for} \quad m = 1,2,...,P \tag{12}$$

where $\tilde{\phi}_{l,m}$ is the binned particle density in grid cell $[l,m]$ and $\lambda = 0,1,,...,P-1$ is the discrete horizontal spatial lag index. Assuming local spatial homogeneity, we then arithmetically average the ACF estimates over all $P$ rows and columns, respectively, to yield two one-dimensional ACF estimates as

$$\widehat{R}^{row}[\lambda] = \frac{1}{P}\sum_{l=1}^{P}\widehat{R}_l^{row}[\lambda] \quad \text{and} \quad \widehat{R}^{col}[\lambda] = \frac{1}{P}\sum_{m=1}^{P}\widehat{R}_m^{col}[\lambda]. \tag{13}$$

We now assume local spatial isotropy and let the arithmetic average of the two perpendicular ACFs serve as a representative single ACF for the adaptation window

$$\widehat{R}[\lambda] = \frac{1}{2}\left(\widehat{R}^{row}[\lambda] + \widehat{R}^{col}[\lambda]\right). \tag{14}$$

Using the estimated ACF, we can finally estimate the local one-dimensional integral length scale $\widehat{L}_c$ for the adaptation window by discretizing Eq. (10) as

$$\widehat{L}_c = \frac{\sum_{\lambda=0}^{P-1}|\widehat{R}[\lambda]|\Delta\lambda}{\widehat{R}[0]}. \tag{15}$$

We now express the correlation length in terms of the associated number of samples as $N_c = \widehat{L}_c/\Delta\lambda$. It is easy to show that $1 \leq N_c \leq P$. We then define the number of effectively uncorrelated particles $N_{\text{eff}}$ as

$$N_{\text{eff}} = \frac{N_g}{N_c}, \tag{16}$$

and it directly follows that $N_g/P \leq N_{\text{eff}} \leq N_g$. The interpretation of $N_{\text{eff}}$ is straightforward: if all particles are spatially uncorrelated, then $N_{\text{eff}} = N_g$, and if all particles are fully correlated (e.g., if they are all trapped in a coherent structure), then $N_{\text{eff}}$ attains its lower limit $N_{\text{eff}} = N_g/P$.

To obtain an estimate $\widehat{\sigma^2}$ of the variance $\sigma^2$ in the two-dimensional binned data, we need to account for the loss of degrees of freedom due to shortening of the residual vector (Bessel's correction) and weighting as well as increased variance due to the binning process itself. The estimate of variance for the binned particle density $\tilde{\phi}_{l,m}$, can then be expressed as

$$\widehat{\sigma^2} = \left(\frac{\sum_{l,m}\tilde{\phi}_{l,m}\|\mathbf{r}_{l,m}-\boldsymbol{\mu}\|_2^2}{\sum_{l,m}\tilde{\phi}_{l,m}}\right)\left(\frac{1}{1-\mathcal{B}}\right) + \sigma_b^2, \tag{17}$$

where $\mathbf{r}_{l,m}$ denotes the grid cell center point position vectors, $\sum_{l,m} \equiv \sum_{l=1}^{P} \sum_{m=1}^{P}$, and

$$\boldsymbol{\mu} = \frac{\sum_{l,m} \tilde{\phi}_{l,m} \mathbf{r}_{l,m}}{\sum_{l,m} \tilde{\phi}_{l,m}} \tag{18}$$

is the weighted mean position vector, and $\left( \frac{1}{1-\mathcal{B}} \right)$, where

$$\mathcal{B} = \frac{\sum_{l,m} \tilde{\phi}_{l,m}^2}{\left( \sum_{l,m} \tilde{\phi}_{l,m} \right)^2}, \tag{19}$$

is a bias correction term that accounts for Bessel's correction and the reduced degrees of freedom due to uneven sample weights (Kish, 1965, pp 86-88). The variance increase due to the binning process (Sheppard's correction, see e.g. Vardeman, 2005) is included through the correction term $\sigma_b^2 = \Delta\lambda^2/12$.

The final local bandwidth estimate (for each particle-containing grid cell) then follows from Eq. (9):

$$\widehat{h} = N_{\text{eff}}^{-\frac{1}{6}} \widehat{\sigma}. \tag{20}$$

### 2.3.3 Boundary solution

We establish a boundary solution for the density estimator by interpolating bathymetry data onto the model grid across all predefined depth layers, creating a matrix of "permissible" and "impermissible" cells for gas. The boundary control is imposed at the kernel estimation stage before summation, by directly modifying kernels whose support contains impermissible cells (see Figure 2). While being computationally intensive, this greatly simplifies the boundary control and entirely omits the difficulties of finding a reliable boundary solution that handles the complex bathymetry and physical processes appropriately.

Impermissible cells are treated as impenetrable obstacles, reflecting that dissolved gas cannot cross land or shallow bathymetric boundaries. Any density within, or "blocked" by impermissible cells, is considered misplaced. A cell is defined as blocked if it lacks a clear line of sight to the kernel center. We determine line of sight using Bresenham's line algorithm (Bresenham, 1965). This is an efficient incremental algorithm relying solely on integer arithmetics that identify grid cells located between an origin cell $(x_0, y_0)$ and a target cell $(x_1, y_1)$ (Figure 2). The algorithm is implemented on a normalized grid with unit cell lengths and initialized by first defining the direction, or step coefficients sx and sy in the $x$ and $y$ directions, respectively:

$$\text{sx} = \begin{cases} 1, & \text{if } x_0 < x_1 \\ -1, & \text{if } x_0 > x_1 \end{cases}, \quad \text{sy} = \begin{cases} 1, & \text{if } y_0 < y_1, \\ -1, & \text{if } y_0 > y_1. \end{cases} \tag{21}$$

and an error term $\varepsilon = \text{dx} - \text{dy}$, where $\text{dx} = |x_1 - x_0|$ and $\text{dy} = |y_1 - y_0|$. The algorithm then iteratively updates $(x_0, y_0)$, using $\varepsilon$ to determine whether to step in $x$ or $y$ direction (sx, sy are not updated) via the following criteria:

$$2\varepsilon > -\text{dy}: \quad x_0 \Rightarrow x_0 + sx \quad \text{and} \quad 2\varepsilon < \text{dx}: \quad y_0 \Rightarrow y_0 + sy. \tag{22}$$

At each time step, $(x_0, y_0)$ is added to the list of grid cells, thereby iteratively forming the line of sight to the target cell. Density in blocked or impermissible cells are redistributed to permissible cells according to the kernel function.

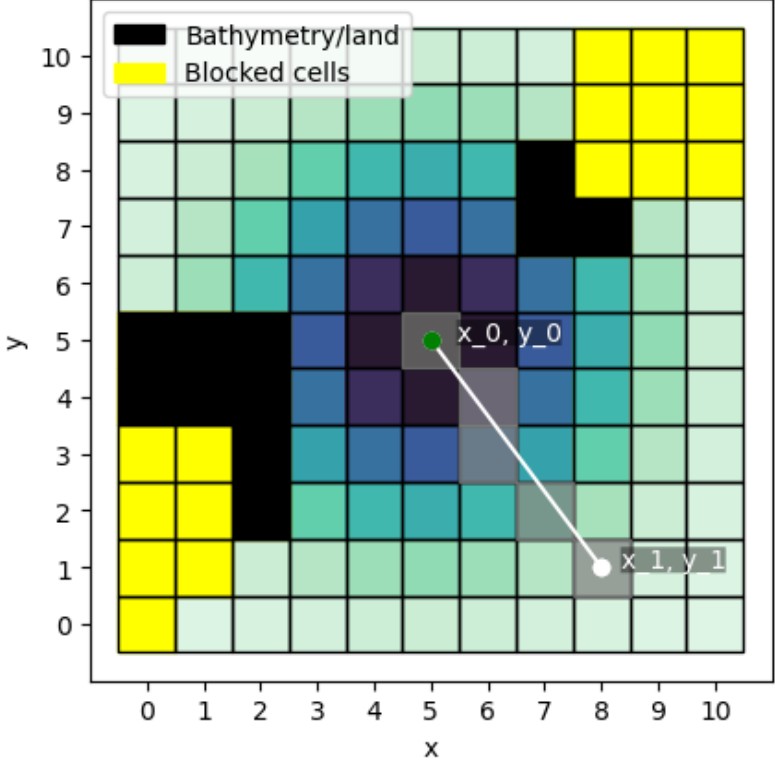

**Figure 2.** Sample kernel with support $i, j = 0, 1, ..., 10$, containing impermissible cells in its kernel support. Density is indicated by blue shading and impermissible cells (Bathymetry/land) and blocked cells where mass from the shown kernel is not permitted to access are shown as black and yellow colored cells, respectively. Cells identified as being in the line of sight between the kernel center (green dot) and cell $[1, 8]$, according to Bresenham's line algorithm, are grayed out (although there is nothing particular about this line).

## 2.4 Atmospheric flux and mass modification functions

Changes in dissolved gas content due to processes within or at the boundaries of the water column are included by modifying the particle masses. Total mass change of a particle $\zeta$ is estimated at each time-step $n$ using predefined mass modification functions that couple particle properties to the gridded field processes. Here, our modeling framework is relatively flexible, and can even accommodate models where it is necessary to keep track of higher order parameters, such as microbe stocks (e.g. a Monod model). This is made possible since we can model any parameter explicitly across the domain. We will only describe mass modification due to dissolved atmospheric exchange of gas here, but a mass modification function for microbial oxidation of $CH_4$ is presented in the application section (Sect 3).

Atmospheric flux can be implemented following any theory using the surface layer concentration as input data. Here we propose a simple solution by applying the bulk equation from Wanninkhof (2014):

$$F = \kappa \left( \Phi_{atm} - \Phi_{sw} \right), \tag{23}$$

where $F$ [mol m$^{-2}$ s$^{-1}$] is the gas flux across the sea-air interface, $\kappa$ [m s$^{-1}$] is the gas transfer velocity, and $\Phi_{atm}$ and $\Phi_{sw}$ [mol m$^{-3}$] are atmospheric and surface water concentrations, respectively. The gas transfer velocity $\kappa$ can be expressed as

$$\kappa(U_a, T_s) = C_\kappa U_a^2 \left( \frac{Sc(T_s)}{660} \right)^{-1/2}, \tag{24}$$

where $Sc(T_s)$ and $C_\kappa$ are empirically derived constants and $U_a$ is the wind speed 10 m above the sea surface. The Schmidt number $Sc(T_s)$ is the empirically derived, gas-specific temperature-dependent ratio between sea water kinematic water viscosity and the diffusion coefficient of the gas. The $C_\kappa$ coefficient lumps together a set of various processes that govern sea/air exchange and has been determined for $CO_2$ and a wind speed range of $4 < U_a < 15$ m s$^{-1}$ using inverse modeling for global estimates. Validity for other gases and wind ranges is not fully known.

Let the gridded estimate of the 2D spatiotemporal atmospheric flux field $\beta(x, y, t)$ be denoted $\widehat{\beta}[i, j, n]$, using the same horizontal and temporal grid cells as $\widehat{\phi}[i, j, k, n]$. We then assume an initial equilibrium between the atmospheric concentration and background surface concentration, which is disturbed by the (modeled) seep-derived dissolved gas. The difference between surface water and atmospheric concentration in Eq. (23) is then simply the surface layer ($k = 0$) concentration $\widehat{\phi}[i, j, 0, n]$. To obtain an estimate of the gas transfer coefficient $\kappa$, we project re-analysis atmospheric 10 meters above sea level wind speed and sea surface temperature data onto all $i, j$, and $n$s, delivering the gridded gas transfer coefficient field estimate $\widehat{\kappa}[i, j, n]$ using Eq. (24). The gridded atmospheric dissolved flux field estimate is then given by

$$\widehat{\beta}[i, j, n] = \widehat{\kappa}[i, j, n] \, \widehat{\phi}[i, j, 0, n] \, \Delta\lambda^2 \Delta t, \tag{25}$$

where $\widehat{\beta}[i, j, n]$ is the integrated atmospheric flux from grid cell $[i, j, n]$.

Loss of gas due to atmospheric flux is implemented by modifying the mass of all particles present in the surface layer. To ensure efficient computation and mass conservation, we assume that the entire contribution to the atmospheric flux from a surface layer particle occurs within the grid cell where that particle resides, disregarding the effects of mass distribution through the density kernels. Errors associated with this assumption are expected to be small, since wind and temperature fields and consequently, gas transfer velocities are generally smooth on typical kernel bandwidth scales. It is also mass conserving, because atmospheric flux varies linearly with dissolved gas concentration (Eq. (25)). Furthermore, since grid cell concentration depends linearly on the total cell gas content (i.e., the sum of all particle masses in that cell) and the gridded gas transfer velocity, relative flux contributions from particles can be estimated using products of particle masses and cell specific gas transfer velocities. The mass loss due to atmospheric exchange for a surface layer particle $\alpha$ at time-step $n$ can then be expressed as

$$\gamma_\alpha[n] = \frac{\Gamma_\alpha[n] \widehat{\kappa}[c(\alpha), n]}{\sum_{\zeta \in \mathcal{A}} \Gamma_\zeta[n] \widehat{\kappa}[c(\zeta), n]} \sum_{i=1}^{I} \sum_{j=1}^{J} \widehat{\beta}[i, j, n], \tag{26}$$

where $\alpha \in \mathcal{A}$ and $\mathcal{A}$ denotes the set of all surface-layer particles, and $c(\alpha)$ denotes the indices $i, j$ where particle $\alpha$ resides.

## 3 Application

We applied the modeling framework to a well documented natural $CH_4$ seep site offshore northwestern Norway located in the Hola trough (Figure 3), where coral reefs and $CH_4$ seeps coexist (Chand et al., 2008). These seeps were investigated not only to assess the mechanisms governing $CH_4$ fluxes to the atmosphere, but also to evaluate their potential impact on cold water coral ecosystems (Sert et al., 2025; Argentino et al., 2025). A thorough description of the data, site characteristics, environmental conditions, and seabed flux estimates are presented in Ferré et al. (2024). In short, the observed seeps are weak, and our focus

is therefore on examining system dynamics and fractional distribution of gas, rather than on quantifying environmental impacts or contributions to the atmospheric $CH_4$ budget.

We modeled the resulting direct and diffusive atmospheric gas release, as well as 3D concentration from 45 observed $CH_4$ seeps for the period between May 20 and June 20, 2018. A 1-month period was chosen since it captures a relatively wide range of periodic variability in both ocean and atmospheric circulation patterns and yields relatively modest computation times. The

OpenDrift simulation required 2-3 days on a supercomputer and the concentration modeling 5-6 hours on a workstation laptop.

### 3.1 Seep gas phase modeling

Free and dissolved gas profiles and direct free atmospheric gas flux were modeled individually for each of the 45 seeps using M2PG1 (see Sect. 2.1) to steady state, using observed and inferred input data and settings, as outlined in the following sections.

#### 3.1.1 Gas phase modeling input data and settings

Temperature and salinity data were extracted from a Conductivity Temperature Depth (CTD) cast performed in May 20, 2018 (Figure 3 and 4a). Seabed gas flux for each seep was estimated using single beam echosounder data (Simrad EK-60 scientific SBE splitbeam echosounder) obtained between May 20 and May 22 2018 and are presented in Ferré et al., 2024. All other input parameters had to be inferred as outlined in the following paragraphs since we lack observations.

M2PG1 requires an initial bubble size distribution, and we used the polynomial fit to visual observations of bubbles as presented in (Veloso et al., 2015). Note that since M2PG1 takes into account the non-spherical shape of bubbles, the bubble size distribution is given using the effective radius $r_E = (a^2b)^{1/3}$, where $a$ and $b$ are the major and minor axis of the spheroid, respectively (Figure 4b). We used the bubble rising speed model from Fan & Tsuchiya, (1990) using their recommendation for bubble contamination, and the linear flatness parametrization from Jansson et al., (2019) (see Figure 4 b and c). We describe

and discuss the bubble rising speed model and deformation parametrization selection in Appendix D.

The horizontal domain size was determined using Appendix A and an assumed barotropic current of $\overline{U} = 0.1$ m s$^{-1}$, horizontal diffusivity $D_h = 0.01$ m$^2$ s$^{-1}$, rising speed $\langle w \rangle = 0.25$ m s$^{-1}$ and a $H = 200$ m deep water column. We estimated $\sigma_w$ using the bubble rise spectrum (Figure 5) to $\sigma_w = 0.025$ m s$^{-1}$. This resulted in an estimated model area of $A_\mathcal{M} \sim 88$ m$^2$ and grid cell side-lengths $9.4$ m. Vertical and temporal resolution does not affect grid cell concentration but must obey the

Courant-Friedrichs-Lewy numerical stability condition. Here we use a grid cell height of $1$ m to obtain $A_\mathcal{M}^\perp = 9.4$ m$^2$ and a

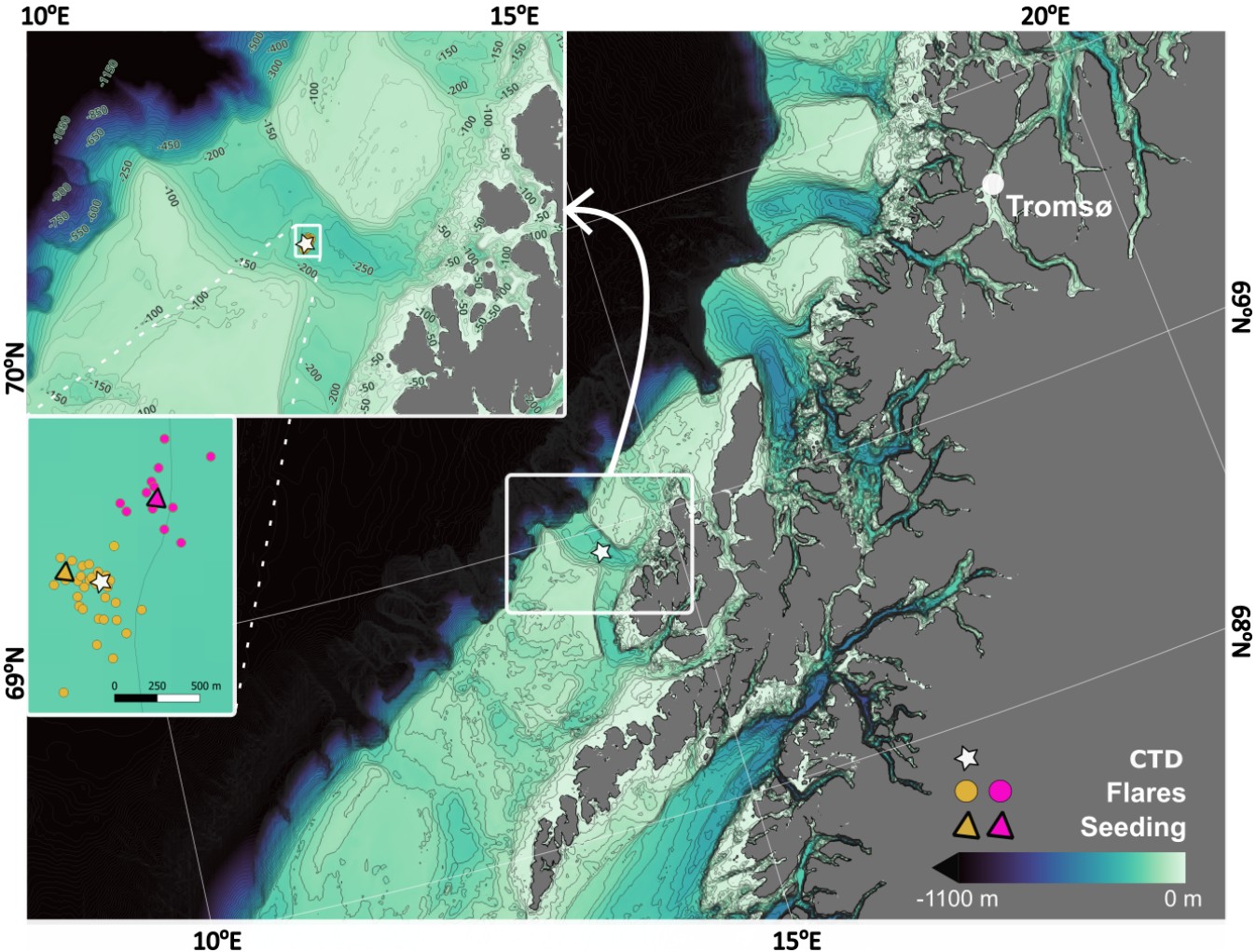

**Figure 3.** Bathymetric map of the application area and location of Conductivity Temperature Depth (CTD) station, observed seep-associated flares indicated by yellow and pink dots during the May 20-22, 2018 survey. Seeding locations (where particles in the particle trajectory model is released), estimated as the flux weighted average position of the incorporated seeps, are indicated by the yellow and pink triangle (see chapter 3.2). Coloring reflects which seeding location each seep observation is pooled into.

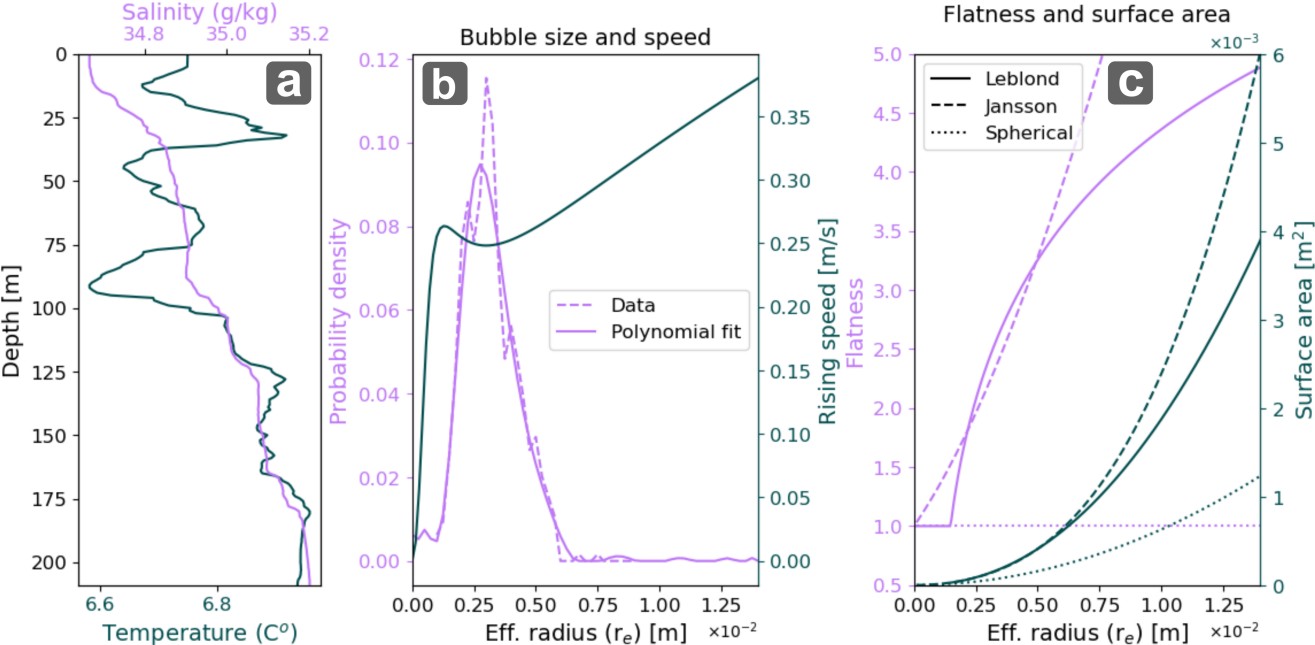

**Figure 4.** Input parameters used in the M2PG1 model runs. a) Conservative temperature and absolute salinity (at the CTD station obtained in May 20, 2018 b) Bubble size distribution (Veloso et al., 2015) and bubble rising speed (Fan and Tsuchiya, 1990) for different bubble sizes as a function of effective radius $r_E = (a^2 b)^{1/3}$ c) Bubble flatness and surface area for various bubble sizes as function of effective radius for Jansson et al. (2019), Leblond et al. (2014), and Spherical flatness parametrization. Note that the linear flatness (Jansson flatness) appears non-linear in the figure since its linearity is with major spheroid axis and not effective radius.

time-step of 0.0625 s. We assumed a constant vertical mixing coefficient of $D_v = 0.001$ m$^2$ s$^{-1}$ and background dissolved gas concentrations were set to the default values from Jansson et al., (2019) at $6.8 \cdot 10^{-4}$ mol l$^{-1}$, $2.5 \cdot 10^{-4}$ mol l$^{-1}$, $2.5 \cdot 10^{-5}$ mol l$^{-1}$, $2 \cdot 10^{-9}$ mol l$^{-1}$ and $1.5 \cdot 10^{-8}$ mol l$^{-1}$ for N$_2$, O$_2$, CO$_2$, CH$_4$, and Ar, respectively. We provide an overview of microbial oxidation rate coefficient ($k_{ox}$) observations presented in literature and their associated uncertainty in Appendix E and Figure 6. Here, we use the simple average $k_{ox} = 3.6 \cdot 10^{-7}$ s$^{-1}$ of the full compiled dataset in Table E1 which include cold seep environments, hydrothermal vents, and human-made releases.

### 3.1.2 Gas phase modeling results

Most of the CH$_4$ gas is dissolved in the water column, with concentration appearing to decrease near exponentially towards the sea surface (Figure 7a). Hourly seabed gas flow rate was $\sim 97$ moles of which 93% dissolved below 100 m depth and only $\sim 0.28\%$ reaching the atmosphere. Integrated atmospheric release from free gas over the 1-month period was 183.1 moles. Free CH$_4$ gas content closely follows the total free gas content throughout the water column (Figure 7a) and loss of total free gas volume (bubble shrinkage, collapse, and dissolution) dominates over other gases replacing CH$_4$ in bubbles. The resulting

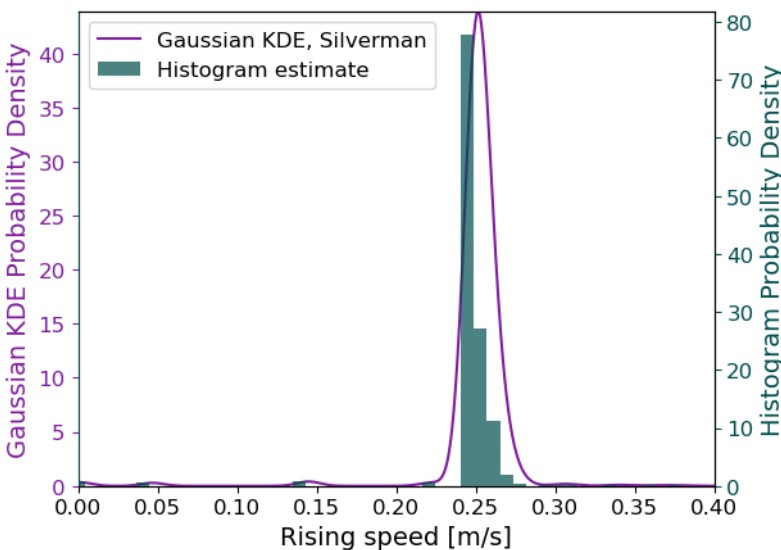

**Figure 5.** Probability density for bubble rising speed of bubbles in the bubble plume using the bubble rising speed model from Fan and Tsuchiya (1990) and bubble size distribution from Veloso et al. (2015).

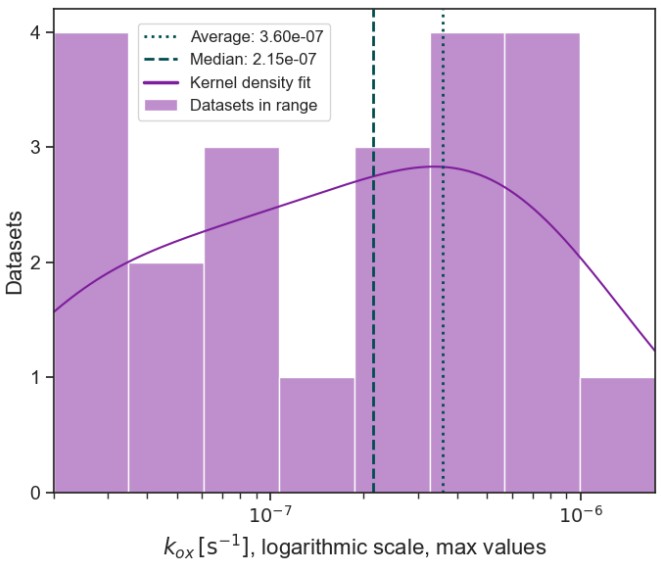

**Figure 6.** Maximum $CH_4$ oxidation rate coefficients ($k_{ox}$) obtained from datasets found in literature and detailed in E and Table E1. The x-axis is logarithmic, meaning that the bars cover different ranges, i.e. the histogram bars are narrower at smaller scales. Vertical dashed lines indicate simple average and median of the values in the table. For the application offshore Northwestern Norway, we used the average of all compiled values.

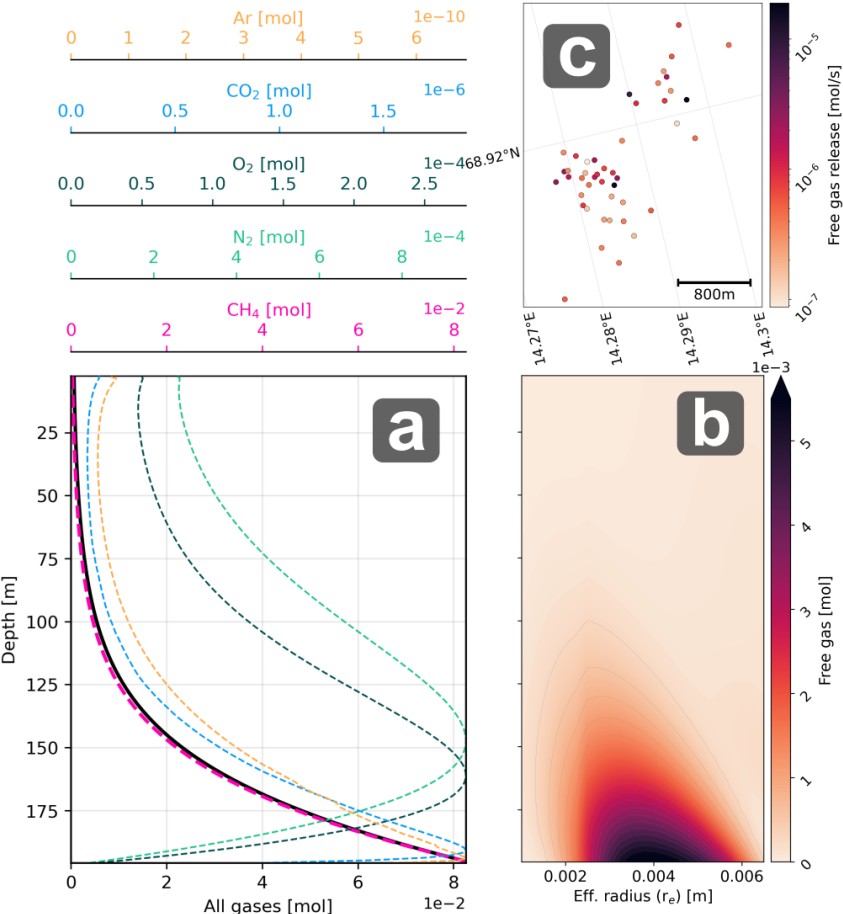

**Figure 7.** a) Vertical profiles of total free gas content for the 5 gases (colored axes) and the total free gas (black, lower x-axis) in the water column for all seeps combined b) Distribution of gas content on bubble sizes for all seeps and gases combined at different depths (note power in scale) c) Free atmospheric gas flux at the sea surface for the 45 seeps (logarithmic color scale).

change in dissolved gas profiles for the four other gases ($N_2$, $O_2$, $CO_2$ and Ar) due to bubble transit, i.e. the transport of gas molecules by entering bubbles, rising, and subsequently dissolving at shallower depths, was therefore negligible (never exceeding $0.1\%$ of background values). Atmospheric flux from the 45 seeps varied considerably from $< 10^{-7}$ to $\sim 10^{-5}$ [mol s$^{-1}$] (Figure 7c), mostly due to large variations in seabed fluxes (Ferré et al., 2024). Dissolved gas injection rates, which are needed as input in the particle dispersion modeling step, were calculated using the modeled (by M2PG1) dissolved gas profiles (not shown) and Eq. (1) and are shown for the 45 seeps in Figure 8a.

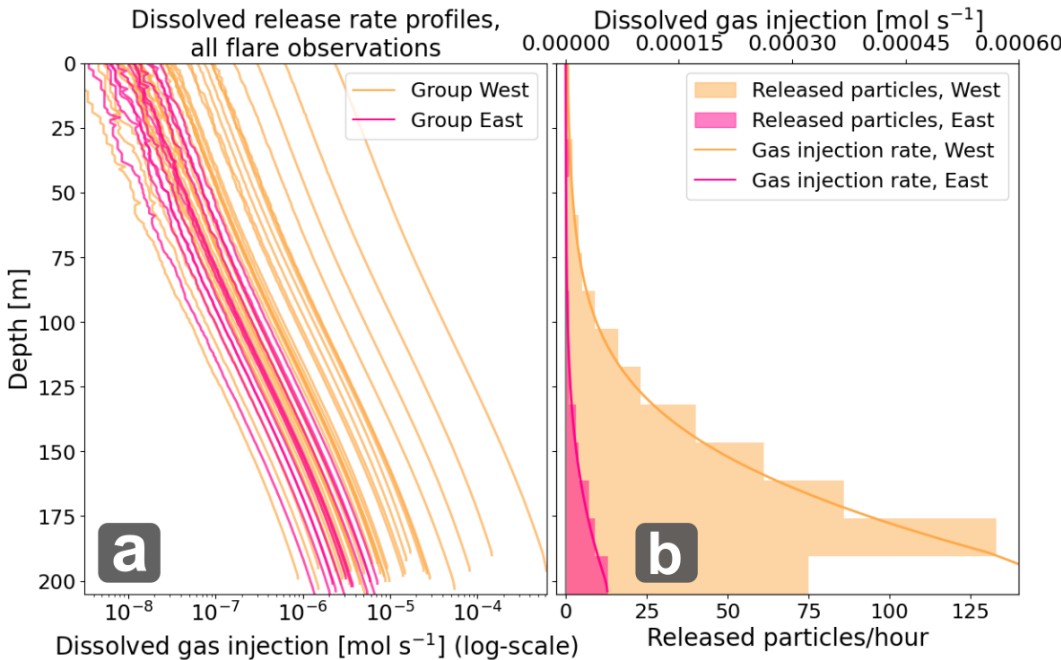

**Figure 8.** a) Dissolved $CH_4$ release rate profiles ($q_m$) for all observed seeps on a logarithmic scale for the two cluster groups (see Figure 3 for seep/group locations, group west in yellow and group East in pink), and b) Resulting accumulated dissolved $CH_4$ release rate from each groups (lines, upper x-axis) and histogram of released particles at each modeling time-step (hourly). The smaller bottom bar at the western cluster reflects the slightly shallower depth in this area.

## 3.2 Particle data set

Using OpenDrift, we simulated a particle data set of $N$ particles with associated 4D-positions (3D space and time) for the 1-month period, as described in Sect. 2.2.

The advective and diffusive components for the drift model (Eq. 2) were determined using velocity vectors and diffusivity coefficients (throughout the water column) obtained from the NorKyst-800 hydrodynamic modeling system. NorKyst-800 is based on the Regional Ocean Modeling System (ROMS, Shchepetkin & McWilliams, (2005)) framework and is a terrain-

following, free-surface, primitive equations model with 35 vertical layers and a horizontal resolution of 800m (Albretsen et al., 2011). The model is eddy-permitting and can resolve major fjord systems and other coastal bathymetric features such as troughs. Vertical turbulent exchange is computed using the General Length Scale closure scheme (Umlauf and Burchard, 2003). OpenDrift has an option to automatically check if diffusivity coefficients are reasonable from a physical perspective, and we used a non-zero fallback value of $D = 0.2$ m$^2$s$^{-1}$ when OpenDrift deemed the input data unphysical. OpenDrift did

not excessively use the fallback value, and from tests where we increased and decreased the value, we found that the choice of fallback value had negligible impact on the results.

We seeded 500 particles every time-step with a particle lifetime of 4 weeks and simulated particle trajectories by updating their positions every 5 minutes. The effect of vertical mixing on the particle trajectory was modeled using a sub-timestep of 30 seconds. We configured OpenDrift to store the particle positions every full hour during the simulation.

The initial particle mass was calculated using seabed gas flux data and Eq. (3). To saturate the particle field prior to the study period, the simulation was initiated on April 20 (1-month spin-up time). With this setup, total particle mass in the domain would increase in the first 10-15 days of the study period due to the re-distribution of removed particle mass. However, particle count would remain approximately constant, with $\Lambda \sim 330\,000$ particles present at each time-step.

Due to the wide range in seep intensity and relatively closely clustered seep positions (Figure 7) combined with the limitation

of 500 release particles each time-step, we chose to aggregate the seeps into two seeding locations to promote smooth release profiles and reduce round-off effects. Grouping was done based on visual inspection of the seep positions (Figure 3) and seed locations were calculated using the flux-weighted average position of each group, given by:

$$\mathbf{r} = \frac{\sum_{g \in \mathcal{G}} \Upsilon_g \mathbf{r}_g}{\sum_{g \in \mathcal{G}} \Upsilon_g}, \tag{27}$$

where $\mathcal{G}$ denotes the set of all seep indices included in the seed position. The seed locations and their associated seeps are

indicated by matching colored triangles and dots in Figure 3. The 500 particles were then distributed according to the injection rate profiles for each seed location with a 1 m vertical resolution, resulting in the profiles/histograms shown in Figure 8b.

### 3.3 Concentration and atmospheric flux estimation

Concentration and atmospheric fluxes were estimated on a $[i, j, k, n]$ grid with cell sizes $\Delta \lambda = 800$ m, $\Delta z = 25$ m, and $\Delta t = 3600$ s, covering the time period between May 20 and June 20 and geographical region between $(12.5^o\text{E}, 68.5^o\text{N})$, $(12.0^o\text{E},$

$72.1^o\text{N})$, $(21^o\text{E}, 72^o\text{N})$, and $(20.1^o\text{E}, 68.45^o\text{N})$. Although particle data were technically available outside of this region, we chose this boundary to avoid potential edge effects in the hydrodynamic model and to constrain computation time. Kernel bandwidths were estimated for each cell location in the 4D grid (each [i,j,k,n]) using a $P \times P$ sized adaptation grid where we determined $P$ using an integral length scale estimate of every 2D $(i, j)$ layer of the particle data. The size of P, typically varying between $\sim 7000$ and $\sim 20000$ m, agreed reasonably well with observations and theory on meso-scale eddy sizes in

the region (Dugstad et al., 2021). Boundary conditions were implemented as described in Sect. 2.3.3 using $[i, j]$- interpolated IBCAO v. 4 bathymetry data (Jakobsson et al., 2012). Spatiotemporal gas transfer velocities $\widehat{\kappa}[i, j, n]$ were estimated from grid interpolated ERA V reanalysis wind and sea surface temperature data (Hersbach et al., 2023) which, together with the surface layer concentration estimates $\widehat{\phi}[i, j, 0, n]$, gave the atmospheric flux field estimates $\widehat{\beta}[i, j, n]$ using Eq. (25).

Particle mass was adjusted at each time-step using the mass modification terms ($\gamma$) for atmospheric flux (Eq. (26)) and

redistribution from removed particles (Eq. (5)). We also added a mass modification term for microbial oxidation, a crucial process when simulating the evolution of dissolved $CH_4$ content in the ocean (Appendix E). We used a simple first order kinetics formula (Eq. E1), with the same rate coefficient $k_{ox} = 3.6 \cdot 10^{-7}$ s$^{-1}$ as in the gas phase model (see Appendix E and Figure 6 for the determination of $k_{ox}$). Microbial oxidation was then included by imposing a mass loss $\gamma_{ox} = k_{ox}\Delta t \Gamma_\zeta[n]$ at each time-step. In principle, this corresponds to discretization of Eq. (E1) using a standard first order forward finite difference

scheme. Mass modification of any particle at any time step could then be calculated by summing up the three applied mass modification terms: i) mass loss to microbial oxidation, ii) mass loss to the atmosphere and iii) mass gained from nearby removed particles.

## 3.4 Application results

The results from the gas phase modeling step is shown and described in Section 3.1.2 and the dissolved concentrations, at-
mospheric flux and fate of $CH_4$ molecules in the modeled domain in the following sections. Animations of the time varying 3D $CH_4$ concentration field and 2D diffusive release field are shown in supplementary material S1 and S2, respectively. It is important to note that the modeling results presented are subject to a wide range of relatively uncertain assumptions concerning various model coefficients and that the main aim here is to test the modeling framework and investigate the dynamics of the system, rather than conclude about absolute values.

### 425 3.4.1 3D $CH_4$ concentration field

The averaged distribution pattern of $CH_4$ throughout the study period is strongly affected by generally northeastward currents that transports gas along the coast, following the shelf and shelf break. The gas enters the more open fjord systems, and to a lesser degree inner fjords. North of $70°$ the $CH_4$ plume disperses more, branching into a northward plume leaving the coast and a coastal plume that keeps following the coastline. The concentration anomaly is generally small, around 2-4 orders of
magnitude lower than typical oceanic $CH_4$ background concentration values ($\sim 3 \cdot 10^{-6}$mol m$^{-3}$, Figure 9), due to the weak seabed release.

The 3D concentration field is very dynamic due to the energetic regional current regimes, and shows variability on ranging from tidal ($\sim$12 hours) to fortnightly periods. A visual representation of the temporal variability of the top 9 layers in the water column (down to 200 m depth) is shown in supplementary material S1 (see video supplement section).
Most of the $CH_4$ is displaced upward relatively quickly from the trough and pushed on top of the shelf break (Figure 9). Vertical distribution of $CH_4$ is therefore characterized by a quick (a few days) shift from $CH_4$ being mainly located close to the seafloor at the release site ($\sim$150-200 m depth) to shallower depths (Figure 10). A thorough analysis of mechanisms causing the rapid shift in location of $CH_4$ is outside the scope of this study, however, upwelling within troughs and along the shelf break is well documented in this region (e.g. Slagstad et al., 1999).

### 440 3.4.2 Diffusive $CH_4$ flux to the atmosphere, microbial oxidation and non-physical redistribution and loss

The time-integrated 2D diffusive atmospheric $CH_4$ release over the study period is shown in Figure 11 and the complete time series can be found in supplementary material S2 (see Video supplement section). Within the model domain, most of the $CH_4$ remains in the water column or is consumed by microbes, with a total of $\sim$0.76% ($\sim$528 moles) being exchanged diffusively with the atmosphere. This diffusive release is roughly three times greater than the local free gas release (0.27%) and does
not account for any diffusive release occurring outside of the model domain. The diffusive flux extends across a broad area

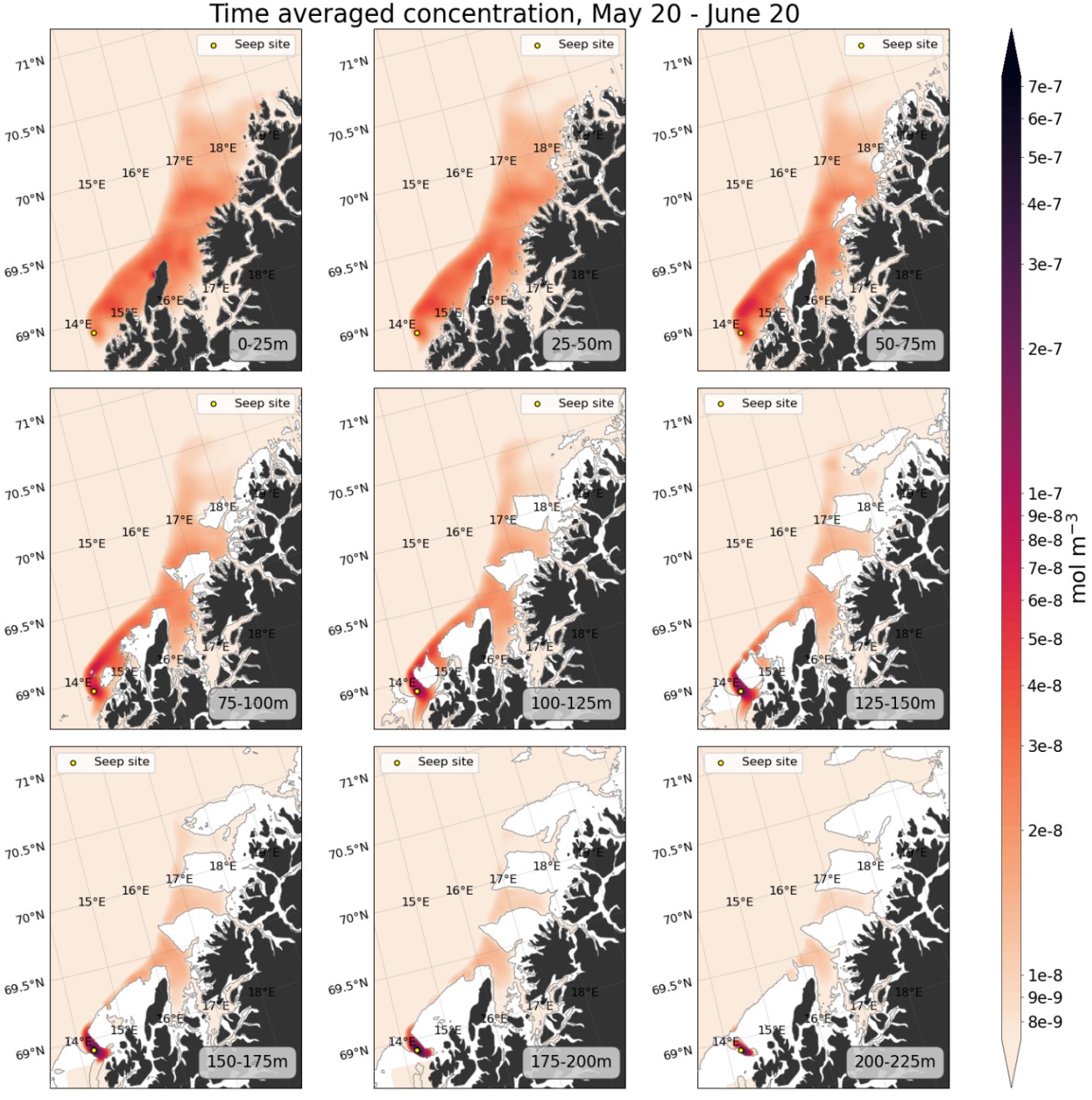

**Figure 9.** Nine depth layers of modification (i.e. the estimate $\widehat{\phi}$ of the anomaly $\Phi'(x,y,z,t)$)) to the CH$_4$ concentration on May 1 as indicated on top of each panel. Typical background concentration in the ocean is $\sim 3 \cdot 10^{-6}$ mol m$^{-3}$ for reference. The bathymetric boundary for the different layers are delineated with a grey contourline.

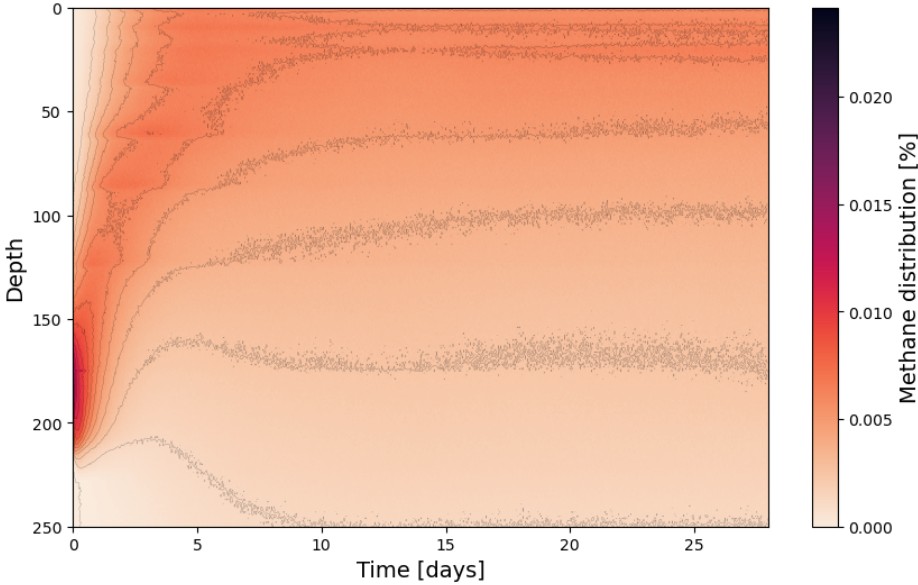

**Figure 10.** Fractional vertical distribution of CH$_4$ in the water column 28 days after release. Color scale shows the fraction of total depth integrated CH$_4$ in the water column.

spanning several hundred kilometers and shows pronounced temporal variability that is strongly correlated with wind speed (Figure 13a), although no clear effect of surface water depletion is observed after storm events. Microbial CH$_4$ consumption exceeds atmospheric flux by a factor one to two orders of magnitude, emphasizing its crucial role in regulating dissolved CH$_4$ levels (Figure 13b). Loss from the domain due to particles leaving the model domain is also substantial ($\sim$5 to $\sim$50 times the atmospheric loss) and shows clear tidal excursions patterns with periodic variability (Figure 13c). Non-physical methane loss due to removed isolated particles (i.e. too far away to be re-distribution) is on the same order of magnitude as loss to the atmosphere via diffusive release (Figure 13d).

### 3.4.3 Fate of released CH$_4$

We analyzed the vertical redistribution and partitioning of CH$_4$ among available sinks within the model domain over a four week period (the particle lifetime). This analysis is important not only for evaluating CH$_4$ molecules potential to reach the atmosphere, but also in cases impacts on water column and/or seafloor ecosystems are of interest. Excluding removed particles and particles leaving the model domain, the accumulated fractional water column CH$_4$ loss due to atmospheric exchange shows an exponential increase the first couple of days, with a subsequent near linear slope until the end of the four week period. The initial rapid gradient increase in atmospheric loss fraction corresponds to a vertical redistribution of dissolved CH$_4$, where the concentration maximum shifts from $\sim$200 m to $\sim$10 m depth (Figure 10). After four weeks, around 0.7% of dissolved CH$_4$ molecules had been transferred to the atmosphere (Figures 12a and b). Microbial oxidation dominates over both atmospheric

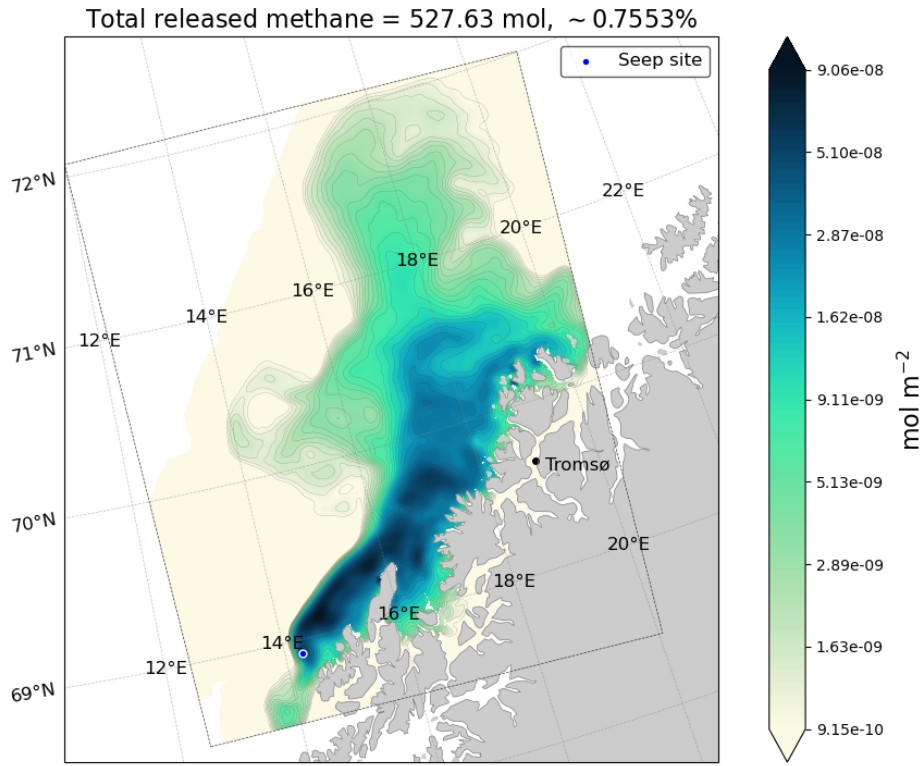

**Figure 11.** Modeled accumulated diffusive release of $CH_4$ within the model domain from the seeps between May 20 and June 20, 2018.

diffusive and free-gas fluxes transforming 65%, while around 34% of the $CH_4$ remains in the water column at the end of the particle lifetime.

### 3.4.4 Interpretation of results

Diffusive exchange of $CH_4$ exceeds the local free gas release and spreads over a large ocean region, making it almost impossible to detect and quantify using conventional measuring instrument. This also poses a challenge for atmospheric inversion models, since these are better at detecting point sources rather than weak releases over large regions (Thompson and Stohl, 2014). These limitations highlight the uncertainty in quantifying the impact of seabed seepage on the atmospheric $CH_4$ budget, particular when considering the potential increased seepage in recent decades due to e.g. thawing marine permafrost, hydrate dissociation 470    (e.g. Serov et al., 2015; Ruppel and Kessler, 2017), and anthropogenic disturbances of the seafloor (e.g. drilling).

    Although here, the estimated total atmospheric flux is small, the impact of more extensive seepage [such as e.g.][](Mau et al., 2017) could be significant and at the same time difficult to observe and/or trace.

    Even though the dissolved gas spreads out over cold water coral reef areas, the $CO_2$ generated by microbial oxidation is likely too small to have any measurable effect on the local ocean environment and cold water corals. This primarily reflects

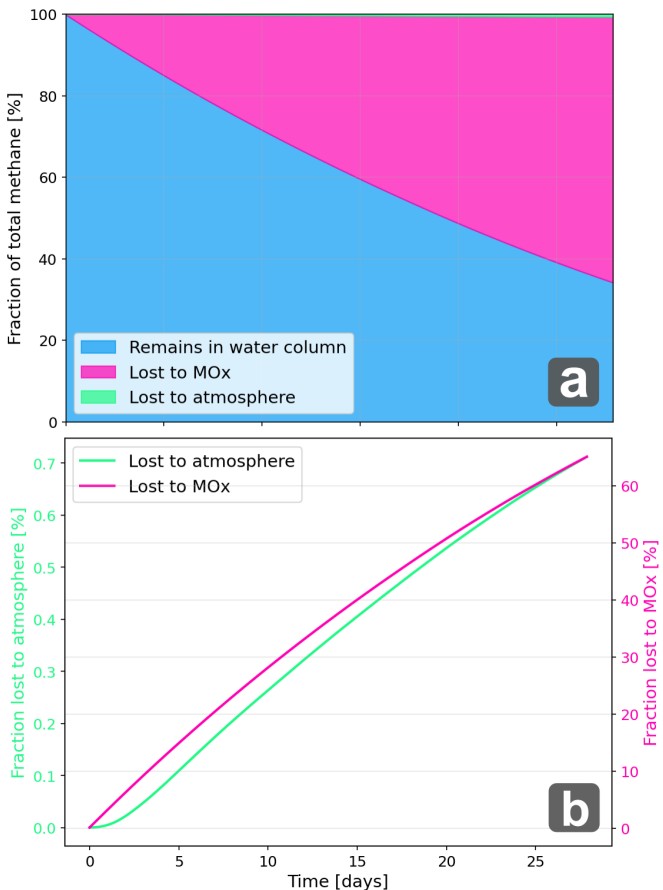

**Figure 12.** a) Accumulated fractional loss to atmospheric exchange and microbial oxidation and fraction CH₄ that remains in the water column and b) Accumulated fractional loss to atmospheric exchange and microbial oxidation in days after release.

the weak seabed fluxes. For more intense and/or localized seepage, e.g. a leaking gas well, this might not be the case. It is also worth noting that the influence of seabed gas seepage on cold-water coral ecosystems remains a sparsely explored field of research.

An additional major caveat, both regarding atmospheric fluxes and potential impact on the ocean environment, is the uncertainty caused by the microbial oxidation rate coefficient $k_{ox}$ assumption in methane oxidation rates (MOx), which vary by
several orders of magnitude and correspond to half-lives for dissolved methane (or methane turnover) ranging from 5 days to nearly 2 years (Figure 6, Table E1). To examine sensitivity to $k_{ox}$, we conducted a limited coefficient-sweep experiment, rerunning the framework using the lowest and highest reported "cold seep" rate coefficients from Table E1: $0.02 \cdot 10^{-6}$ s$^{-1}$ (low) and $0.98 \cdot 10^{-6}$ s$^{-1}$ (high) (Gründger et al., 2021), as well as two intermediate values.

The low and high rate coefficient runs resulted in a ~34% increase (705 moles) and ~41% decrease (309 moles), respec-
tively, in atmospheric emissions during over the model domain and particle lifetime. The impact on the final fate of dissolved

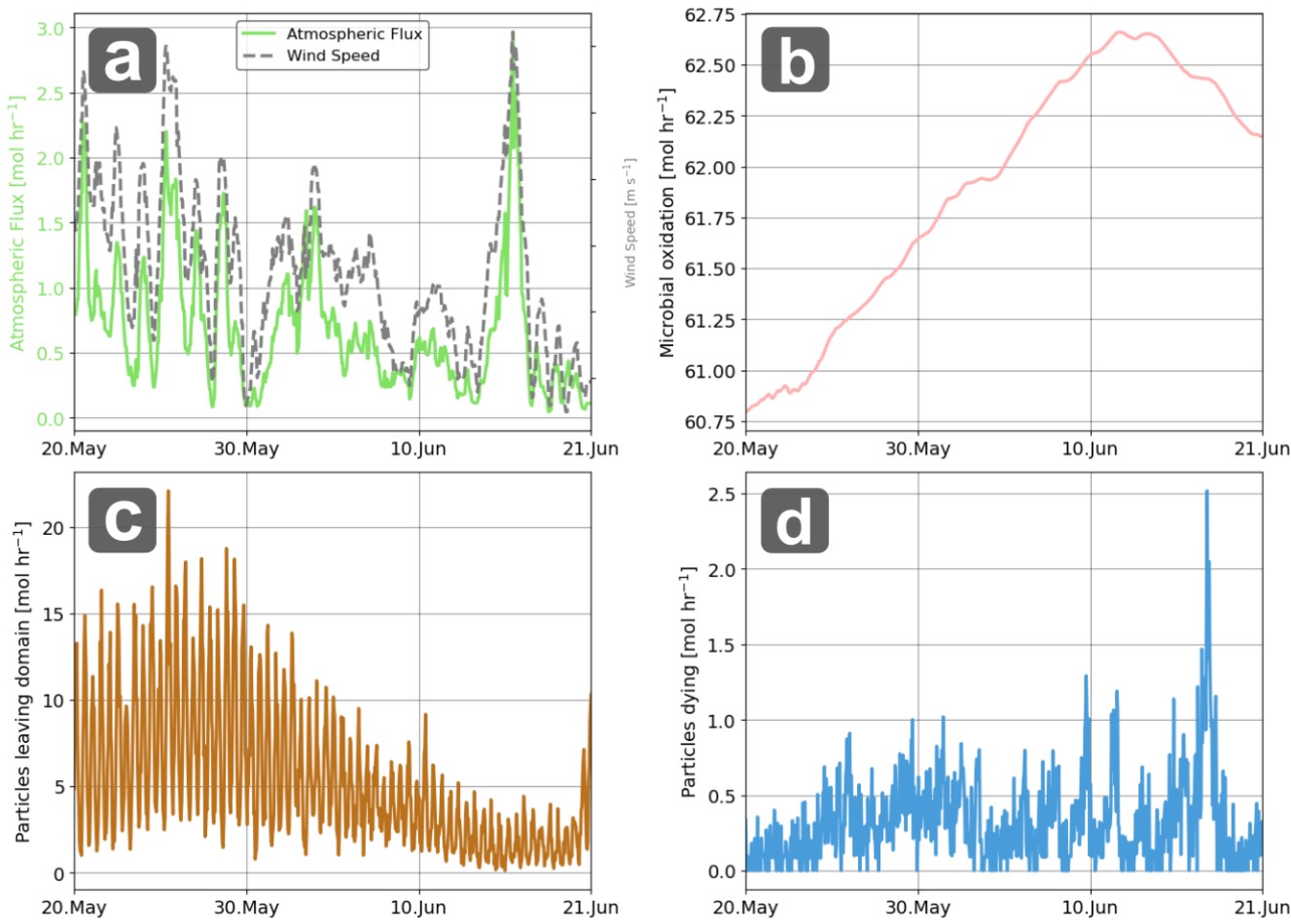

**Figure 13.** a) Loss of $CH_4$ from the water column from atmospheric equilibrium, b) Microbial oxidation, c) Particles leaving the model domain and d) Mass loss due to deactivation of particles that are unable to redistribute its mass.

$CH_4$ molecules were also considerable (Table 1). After 4 weeks, $\sim$5.5% of dissolved $CH_4$ were consumed and 1.16% released to the atmosphere in the low rate coefficient run, compared to $\sim$91% consumed and 0.36% released in the high rate coefficient run. These results highlights the importance of selecting a suitable MOx rate coefficient and illustrates the huge span of results one can obtain when coefficients in the modeling framework are poorly constrained. It is also important to note that since MOx
is a biologically mediated process, it can vary substantially on relatively small spatiotemporal scales (Valentine et al., 2001; Ruff et al., 2015) depending on a wide range of factors, including $CH_4$ concentration, water temperature, salinity (Steinle et al., 2015; Osudar et al., 2015), nutrient availability (Knief, 2015), and the presence of trace elements (Hanson and Hanson, 1996). Thus, assuming a constant rate coefficient is in itself a potentially problematic simplification, since it will most likely vary considerably across the model domain. One would, for instance, expect MOx to decrease with the distance from seep area
due to rapid dilution of methane and varying environmental stress for the $CH_4$ oxidizing microbes. To reduce the uncertainty

concerning MOx rates, future studies must therefore not only constrain rate coefficients, but also improve our ability to model MOx dynamics within modeling frameworks. Including more complex MOx parametrizations is possible in our framework since we allow explicit modeling of higher order fields at each time-step and location in the modeling domain.

| $k_{ox}$ ($10^{-6}$ s$^{-1}$) | 0.02 | 0.18 | 0.36 | 0.67 | 0.98 |
|---|---|---|---|---|---|
| MOx (%) | 5.5 | 31.3 | 65.2 | 83.7 | 91.4 |
| Atmospheric release (%) | 1.2 | 1.1 | 0.7 | 0.5 | 0.36 |
| Remains in WC (%) | 93.3 | 67.6 | 34.1 | 15.8 | 8.2 |
| Diffusive release (mol) | 705 | 640 | 528 | 392 | 309 |

**Table 1.** Fate of CH$_4$ 4 weeks after it is dissolved in water using a sweep of different microbial oxidation rate coefficients ($k_{ox}$).

Another important source for uncertainty in the modeled fate of CH$_4$ arise from the atmospheric bulk model. The atmospheric gas transfer coefficient function (Eq. (24)) was derived based on, and designed for global CO$_2$ estimates (Wanninkhof, 2014) and has an estimated uncertainty of $\sim 20\%$, even for its intended use. We must expect considerably higher uncertainties due to our application in a local coastal region (as opposed to an ocean basin), where wind speeds may exceed the validity range, and since the specific coefficient $C_\kappa$ has been determined solely for CO$_2$.

Uncertainties in the eddy diffusivity, vertical transport and distribution are also expected to be large. The choice of grid cell thickness can also modify the end result. If the grid cells are too thin, and temporal resolution too coarse, there is a risk of depletion of the surface layer between the model output time-steps. On the other hand, if the grid cells are too thick, one would incorporate CH$_4$ from depths where exchange with the atmosphere is unrealistic, thereby violating the assumptions of the atmospheric exchange bulk model. One can evaluate whether the surface thickness is sufficiently thick by comparing typical values for Eq. 26 with the typical mass of surface layer particles and ensure that the atmospheric loss is considerably smaller than the surface layer gas content (i.e. that $\gamma_\alpha[n] << \Gamma_\tau[n]$ always).

## 4   Conclusions

We implemented and applied a new framework for modeling the impact of seabed gas seepage on spatiotemporal water column concentrations and atmospheric gas exchange with the ocean. The application uncovered and highlighted important aspects of the dynamics regarding the fate of seeped gas from the seabed, such as a highly distributed diffusive release which considerably exceeds local free atmospheric gas fluxes.

Estimation uncertainties arise from a relatively wide range of sources which should be addressed in future studies. In particular, mass loss due to microbial oxidation pose a significant challenge since rate coefficients are shown to exhibit large variability which cause considerable differences in the modeled atmospheric fluxes and concentrations. Current parameterizations of mass loss due to atmospheric ventilation are also simple and developed for large scale ocean regions, not coastal areas. Our results are also sensitive to poorly resolved diffusivity coefficients, which can greatly affect the water column distribution of dissolved

gas. A steady state assumption for the seepage itself might also be a problematic assumption in areas where seepage is known to vary strongly over time (e.g. Ferre et al., 2020).

From a pure modeling perspective, non-physical re-distribution of dissolved gas is also a source for potential error and when estimating the final fate of gas, the limitation in model domain size makes inference about final state difficult. In cases where the aim is to estimate the final fate of released gas, one could argue that it is more suitable to use a 1D approach, e.g. as suggested in Nordam et al., (2025). Using a steady state solution of the gas phase model is also a drawback and might currently cause significant errors, especially for intense seep sites where background gas concentrations can be significantly altered.

On the other hand, the framework is flexible and reasonably fast and makes it possible to employ complex, locally adapted existing hydrodynamic models and can include advanced process modules, thereby capturing not only idealized processes but also complex hydrodynamic and chemical/biological phenomena. It has a wide range of potential applications, not only for monitoring known gas seeps, but also for risk assessments concerning future potential increased seepage due to e.g. hydrate thawing, James et al., 2016, leaking gas wells, and integrity of subsea legacy carbon storage reservoirs (e.g. Torsæter et al., 2024) and other leaking industrial installations. Certain studies also requires a 3D spatiotemporal concentration field, e.g. when studying the potential effect of seepage on biological processes in a specific area. Aspects of the framework (e.g. the kernel density estimator) can also be complementary to established frameworks for post-processing and analysis of ocean particle dispersion data and contaminant spreading in the ocean in general (e.g. the ChemicalDrift module, Aghito et al., 2023).

Aside from improving process and atmospheric exchange modules, future developments should consider a full on-line coupling between concentration model and gas phase model as well as a proper validation study to ensure realistic results.

*Code and data availability.* Code for creating input data to and output data from M2PG1 as well as running the model in batch for multiple seeps is freely available at DOI: 10.5281/zenodo.15042452 or GitHub. The adaptive kernel density estimator and testing of the adaptive kernel density estimator can be accessed at DOI: 10.5281/zenodo.15042426 or GitHub and the code for the whole framework, as well as seed profiles, including the code used for running OpenDrift can be accessed at DOI: 10.5281/zenodo.15042437 or at GitHub. The particle position output data from the OpenDrift model run can be accessed at DOI: 10.5281/zenodo.15042308.

*Video supplement.* Supplementary material 1 (SI1) can be accessed at DOI: 10.5446/69942 and supplementary material 2 (SI2) can be accessed at DOI 10.5446/69941.

## Appendix A: M2PG1 Model Grid Cell Dimensions

Here we propose a solution for determining a reasonable horizontal model grid cell size assumption for M2PG1, as no established method currently exist. Since M2PG1 assumes horizontally invariant concentrations within the predefined model domain, the choice of the horizontal model domain size directly affects the concentration within the model grid cells and, in turn, gas transfer and dissolution. Defining the horizontal dimensions of the model grid cells must therefore be done with care

and should reflect the horizontal extent of the modeled bubble plume to obtain realistic results. We determine the horizontal and vertical gas phase model grid cell area, respectively denoted $A_\mathcal{M}$ and $A_\mathcal{M}^\perp$, by modeling the 2-dimensional spread of the bubble cloud.

We assume that the seeps are point sources and that the bubbles drift with a barotropic current with mean speed $\overline{U}$ and random velocity fluctuations governed by a horizontal diffusivity $D_h$. In this framework, horizontal bubble spread is caused by i) differences in accumulated horizontal displacement resulting from varying rising speeds of bubbles with different sizes (slow/fast bubbles spend more/less time in the velocity field) and ii) turbulent (random) effects in the horizontal flow, modeled as diffusion. The horizontal extent of the bubble plume increases towards the sea surface and we use the estimated spread at half of the total water column depth $H$ to minimize estimation errors at the surface/bottom.

The spread due to differences in rising speed can be estimated using the probability density $P$ for bubble rising speeds $w_o$ in the bubble cloud. The distribution of rising speeds for bubbles in the bubble cloud can be described by the discrete probability $P[w_o]$, and can be derived from the chosen initial discrete bubble size distribution (BSD) and bubble rising speed model. This is done by estimating the bubble rising speed of all bubbles in the BSD and re-bin the results, using the fractional weights from the BSD, according to bubble sizes. We obtain the weighted distribution average and standard deviation as

$$\langle w \rangle = \frac{\sum_o w_o P[w_o]}{\sum_o P[w_o]} \quad \text{and} \quad \sigma_w = \sqrt{\frac{\sum_o (w_o - \langle w \rangle)^2 P[w_o]}{\sum_o P[w_o]}}, \tag{A1}$$

where $w_o$ are discrete rising speeds, $P[w[o]]$ associated probabilities, $\langle w \rangle$ the weighted average, and $\sigma_w$ weighted the standard deviation (see Figure 5 for an example). Note that the BSD is expected to change with height above the seafloor (which also changes $P[w_o]$). For the purpose of this calculation, however, we assume the BSD remains unchanged.

Along-flow spread $\Delta x_{rs}$ can then be expressed as

$$\Delta x_{rs} = \overline{U} \Delta t_{max}, \quad \text{where} \quad \Delta t_{max} = \frac{H}{2} \left[ \frac{1}{\langle w \rangle - \sigma_w} - \frac{1}{\langle w \rangle + \sigma_w} \right]. \tag{A2}$$

Horizontal displacement due to current diffusivity acts in both along-flow ($x$) and cross-flow ($y$) direction and can be expressed by the 2D Gaussian solution to the diffusion equation for a point source,

$$p(x,y,t) = \frac{1}{\sqrt{4\pi D_h t}} e^{-(x^2+y^2)/4D_h t}, \tag{A3}$$

where $p(x,y,t)$ is the normalized count of bubbles at position $(x,y)$ and time $t$ and $4D_h t$ is the variance of the spread in both directions. We constrain diffusive spread using twice the standard deviation $2\sigma_D$ of the distribution at $H = H/2$ given by

$$\Delta x_D = \Delta y_D = 2\sqrt{2D_h} \cdot \sqrt{0.5 t_H} \quad \text{where} \quad t_H = \frac{H}{\langle w \rangle}. \tag{A4}$$

and

$$A_\mathcal{M} = (\Delta x_{rs} + \Delta x_D)\Delta y_D \tag{A5}$$

giving horizontal grid cell side lengths of $\sqrt{A_\mathcal{M}}$ (since M2PG1 uses square cells).

An estimate of the vertical grid cell area, which is needed to estimate dissolved gas injection profiles is easily obtained and defined as

$$A_{\mathcal{M}}^{\perp} = \sqrt{A_{\mathcal{M}}} \Delta z_{\mathcal{M}}, \tag{A6}$$

where $z_{\mathcal{M}}$ is the vertical grid cell size.

## Appendix B: The histogram estimator

A commonly used density estimator based on data from particle dispersion models is the histogram estimator. The histogram estimator for the concentration estimate $\hat{\phi}$ at position $\boldsymbol{r}_0$ using a predefined grid with grid cell volume $V$ can be expressed as

$$\hat{\phi}(\boldsymbol{r}_0) = \frac{1}{V} \sum_{\zeta=1}^{Z} \Gamma_\zeta K(\boldsymbol{r}_0, \boldsymbol{\eta}_\zeta), \tag{B1}$$

where $\Gamma_\zeta$ and $\boldsymbol{\eta}_\zeta$ represent the mass and positions (respectively) of particles, and

$$K(\boldsymbol{r}_0, \boldsymbol{\eta}_\zeta) = \begin{cases} 1 & \text{when } \boldsymbol{\eta}_\zeta \text{ shares the same grid cell as } \boldsymbol{r}_0, \\ 0 & \text{otherwise.} \end{cases} \tag{B2}$$

Using the histogram estimator implies modeling a smooth, continuously distributed property with a discontinuous, quantized, and piece-wise constant function, which introduces several drawbacks with this estimator-property pairing. Firstly, the estimator is highly dependent on the choice of grid cell size: fine grids result in noisy and unrealistic estimates in regions with medium to low particle counts, while coarse grids lead to significant loss of information in areas with high particle counts. Secondly, the histogram estimator can be sensitive to the chosen position of the origin. In addition, the minimum concentration

estimate is limited to one particle per grid cell, which can significantly influence e.g. atmospheric flux estimates (for instance if that concentration exceeds the atmospheric background concentration). Some of these issues can be mitigated by adjusting the grid cell size, however, the problems prevail in highly heterogeneous domains containing regions with low particle saturation (unless one adobts an unstructured grid). Seeding more particles is always a remedy, however, we are still left with inefficient use of the particle position data and potentially unfeasible computational complexity (see Sect. 2.2.2). We have therefore for-

mulated an adaptive bandwidth, 2D grid-projected Kernel Density Estimator (KDE) specifically for OpenDrift output data to calculate the concentration field.

## Appendix C: Density estimator testing and validation

The adaptive kernel density estimator was developed, tested, and compared with other estimators using a numerical toy model that generates data resembling typical OpenDrift data. The toy model gives full control of all parameters and allows to effi-

605 ciently test various scenarios due to the low computational cost of each run. Here we compare the adaptive bandwidth KDE against other estimators as explained below.

## C1  Toy model and test simulation

The synthetic data was designed to mimic output data from OpenDrift by seeding particles at seed location $\mathbf{r}_0$ and calculating their position at time step T (at time $T\Delta t$) as

$$\mathbf{r}_T = \mathbf{r}_0 + \sum_{\vartheta=0}^{T-1} \left( \mathbf{U}_\vartheta \Delta t + \boldsymbol{\xi}_\vartheta \sqrt{2D\Delta t} \right), \tag{C1}$$

where $\mathbf{U}_\vartheta = [u_\vartheta, v_\vartheta]$ represents a spatially uniform, time varying velocity field, $\vartheta = 0, 1, 2, 3, ... T$ and $\Delta t$ is the modeling time-step, $\boldsymbol{\xi}_\vartheta = [\xi_{\vartheta,x}, \xi_{\vartheta,y}]$ where each component is sampled from a standard normal distribution $\mathcal{N}(0,1)$ and $D$ represents a diffusivity coefficient (assuming isotropy). The velocity $\mathbf{U}_\vartheta$ was calculated as

$$\mathbf{U}_\vartheta = \frac{\mathbf{U_0} + \boldsymbol{\Xi}_\vartheta}{\|\mathbf{U_0} + \boldsymbol{\Xi}_\vartheta\|} \|\mathbf{U_0}\| \tag{C2}$$

where $\mathbf{U_0} = (u_0, v_0)$ gives the initial velocity, $\boldsymbol{\Xi}_\vartheta = (\sin[\frac{\vartheta}{50}]u_0, v_0)$ and $\|\cdot\|$ gives the euclidean norm. The normalization with $\|\mathbf{U_0}\|$ is necessary to ensure conservation of mass in the field. We choose $U_0 = (0.1, 0)$ and $D = 0.14$ and released a total of $2 \cdot 10^6$ particles from a point source at $\mathbf{r}_0 = (10, 10)$ over the course of 400 timesteps. The histogram density estimate (Eq. B1) of the full simulation was considered the simulation "Ground-truth" and is shown in Figure C1. The computation time for generating the test data was 8.3 seconds with a Intel Core Ultra 9 185H processor.

## C2  Testing and evaluation of different estimators

We implemented and tested four estimators i) The histogram estimator, ii) An Time-dependent bandwidth estimator, iii) The Silverman bandwidth estimator from the *gaussian_kde* function from the *scipy.kde* python package, and iv) The adaptive bandwidth estimator used in the present study. All estimators were tested on a data set where we picked every 1000th particle from the full data set of $2 \cdot 10^6$ resulting in a total of 2000 particles for density estimation. All estimates were done grid-projected as described in Sect. 2.3.1 and for the final time-step only and all particles had a mass of 1.

For the histogram estimator estimate, we used Eq. (B1) and for the time-varying bandwidth estimator, we defined the bandwidth as

$$h_{tv} = \sqrt{4Dt_\vartheta} \tag{C3}$$

which is the theoretically ideal bandwidth for the time-varying estimate. In a real-world scenario, the diffusion coefficient varies and we cannot estimate the correct diffusion coefficient unless information about the local diffusivity is given from the hydrodynamic model. Although the bandwidth function could be suitable when such information is available, complex bathymetry may introduce challenges as discussed below. For the estimate using the provided *scipy.kde.gaussian_kde*, we used default settings and the *bw_method='silverman'* setting (SciPy Community, 2024). We refer to the package documentation for details (URL in the reference list). For the adaptive bandwidth estimator, we followed the algorithm described in Sect. 2.3.2 and estimated $h$ locally for each particle-containing grid cell. For the "in-house" coded estimators (all but *scipy.kde.gaussian*) we included the boundary control explained in Section 2.3.3.

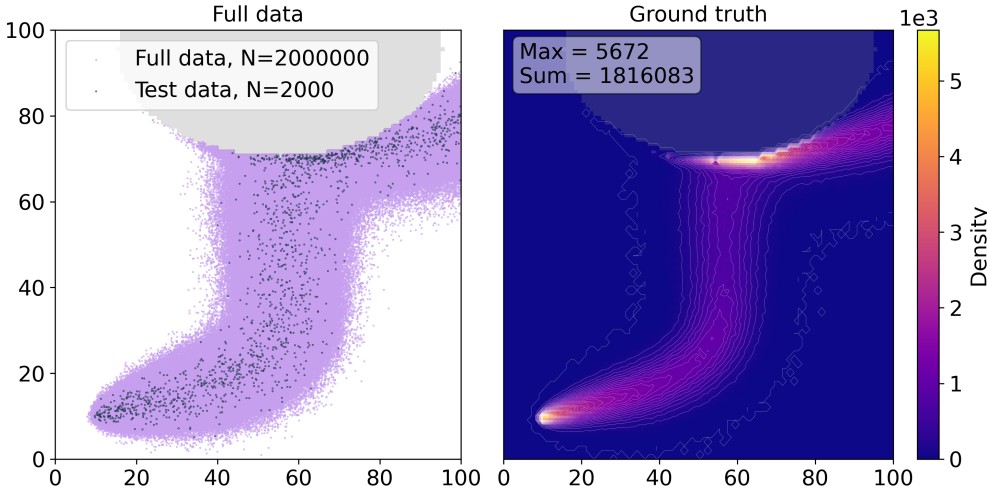

**Figure C1.** Left: Synthetically generated particle dispersion model data for $2 \cdot 10^6$ particles (purple) and randomly picked "test data" of 2000 particles (green dots) in a domain with a simple "impermissible" elliptic boundary. Right: Histogram estimate of the full $2 \cdot 10^6$ particle dataset, representing the "ground truth" in the test scenario for the adaptive kernel density estimator.

A comparison of the four estimators and a residual analysis plot are shown in Figure C2. In addition to a visual comparison, we evaluate how the max values in the field aligns and do a simple $R^2$ statistic. The Histogram estimator gives a noisy result, with a very high max value of 9000 compared to 5672 for the ground truth and a low $R^2 = 0.53$. The the non-adaptive Silverman results in an unrealistically smooth estimate, with a very low maximum value of 927 and low $R^2 = 0.59$. While the time varying bandwidth estimator works relatively well in the open "unbounded" part of the domain, it over-smooths when encountering the boundary - highlighting a problem with time varying bandwidth in bounded domains where the stochastic process is limited by physical obstacles. Nonetheless, it performs better than the non-adaptive Silverman and Histogram estimators, achieving an $R^2 = 0.71$. The adaptive bandwidth estimator is in general slightly over-smooth, however, it significantly outperforms the three other estimators with a maximum value of 4531, which is the closest to the ground truth of 5672 and has a high $R^2 = 0.90$ (Figure C2).

The total computation time for doing all the KDE estimates (including the kernel adaptation) were less than 1 second with a Intel Core Ultra 9 185H processor, and the adaptive kernel density estimator was only slightly slower than the *gaussian_kde* function from the *scipy.kde* package (both $\leq 10^{-3}$). A simple comparative performance study as well as script for further testing and evaluation of the adaptive kernel density estimator and the estimators used for comparison is available at the GitHub repository linked in the assets section of this article ("Code for the adaptive kernel density estimator").

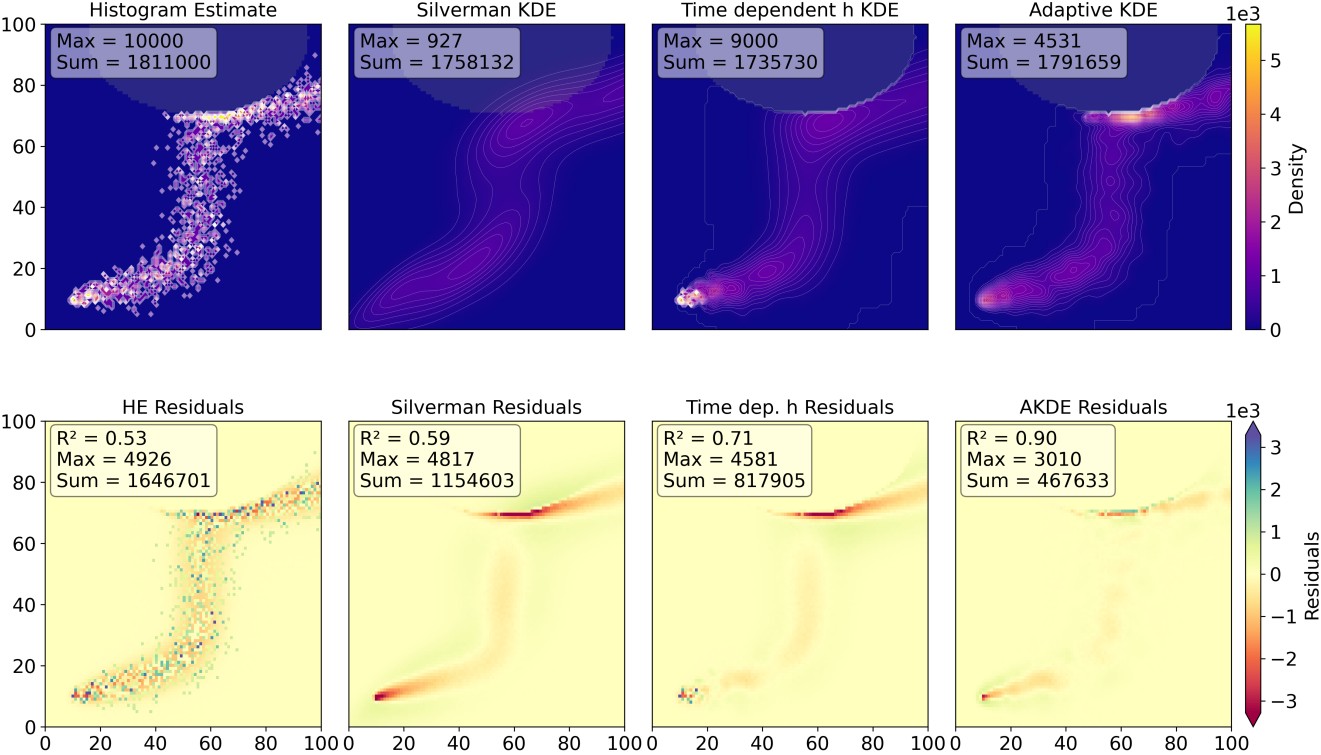

**Figure C2.** Density estimates using the test dataset (N=2000, green dots in Figure C1) using, from left to right, a histogram estimator, a Silverman (non-adaptive) bandwidth estimator from the *scipy.kde* package, a Time-dependent bandwidth kernel density estimator (TKDE) with boundary control, and the Adaptive bandwidth kernel density estimator (AKDE) developed here. The upper panel figures show the density estimates and the lower panel figures the residuals from the ground truth estimate shown in Figure C1. The impermissible region (land/bathymetry) is shaded in grey.

## Appendix D: Rising speed model and flatness parametrization

It is well-known that the terminal bubble rise veloocity $U_b$ varies non-linearly with the bubble size (e.g., (Fan and Tsuchiya, 1990), (Leifer and Patro, 2002)). In addition, it depends on fluid and gas parameters, and the degree of contamination. Fan and Tsuchiya (1990, Eq. (2.11)) showed that the terminal bubble rise velocity can be written as

$$U_b = \left( U_{b1}^{-c} + U_{b2}^{-c} \right)^{-1/c}, \tag{D1}$$

where $U_{b1}$ dominates for small bubbles, and $U_{b2}$ dominates for large bubbles. The dimensionless parameter $c$ (called $n$ in (Fan and Tsuchiya, 1990) and $d$ in (Leifer and Patro, 2002)) is a measure of the degree of contamination, which directly affects the surface tension of the bubbles. By fitting Eq. (D1) to multiple experimental data sets, Fan and Tsuchiya (1990) found that $0.8 \leq c \leq 1.6$, where the lower limit corresponds to contaminated bubbles, and the upper limits corresponds to clean bubbles.

We follow their recommendation, and apply $c = 1.2$ for our assumed moderately contaminated conditions, see Figure 4b and Jansson et al., (2019) for further details.

Bubble deformation is an important factor in bubble dissolution and exchange rates of gas since it changes the surface area to volume ratio of the bubbles. Deformation can be characterized by a dimensionless flatness ratio, defined as $f \equiv a/b$. In addition to spherical flatness, two parametrization options are available in M2PG1: Leblond flatness (Leblond et al., 2014), where $f = 0.45 + 1.4 \ln(a/b_{ref})$ for $a > 1.48$ and Jansson flatness (Jansson et al., 2019), where $f = 1 + 0.3064(a/b_{ref})$ (coined here, referred to as "linear flatness" in Jansson et al., 2019) and $b_{ref} = 1$ mm. While Jansson flatness parametrization has support for bubbles where $a < 1.48$ mm, it lacks an empirical basis other than a fair agreement with Leblond for relatively small bubbles. The divergence between the two models at larger bubble sizes (Figure 4c) can lead to misrepresentations when modeling distributions that are skewed towards larger bubbles. Nonetheless, we use Jansson flatness parametrization since here our observations indicate that most of the gas is confined to smaller bubble sizes (Ferré et al., 2024).

## Appendix E:  Brief review of existing estimates of MOx rate coefficients

Oxidation of $CH_4$ to carbon dioxide is achieved by several groups of aerobic methanotrophs and reaction rates vary substantially, depending on existing microbial consortia, stoichiometry of the involving nutrients, and overall succession of the methanotrophs (Hanson & Hanson, 1996). Nonetheless, reaction rate measurements with radiotracer assays highlight first-order reaction kinetics and a $CH_4$ decay rate following

$$\frac{d\phi(t)}{dt} = -k_{ox}\phi(t) \quad \Rightarrow \quad \phi(t) = \phi_0 e^{-k_{ox}t} \tag{E1}$$

where $k_{ox}$ is the reaction rate and $\phi_0 = \phi(0)$ the initial concentration. Most measurements of microbial $CH_4$ oxidation (MOx) in marine environments are focused on locations where $CH_4$ concentrations exceed the background levels. The rates of $CH_4$ oxidation in suboxic zones, hydrothermal vents, and cold seeps exhibit substantial variability, spanning several orders of magnitude from $10^{-8}$ to $10^{-2}$ nM s$^{-1}$, primarily due to spatiotemporal fluctuations in $CH_4$ concentrations. In contrast, half-life or $CH_4$ oxidation rate constants ($k_{ox}$) are independent of $CH_4$ concentration and provide a more accurate representation of the water column's MOx capacity. The rate coefficients ($k_{ox}$) range from $0.02 \cdot 10^{-6}$ s$^{-1}$ to $1.74 \cdot 10^{-6}$ s$^{-1}$, corresponding to halving times of approximately five days to two years. However, $CH_4$ can remain stable for decades in oxygen-limited environments where aerobic $CH_4$ oxidation is inhibited.

*Author contributions.*  Writing was done by KOD, HE, AH, MFS, and BF. Data were curated by KOD, HE, and BF. Original draft preparation and software development were done by KOD. Method development was done by KOD, HE, and AH. Investigation and formal analysis were done by KOD, HE, AH, MFS, and BF. Visualization was done by KOD. The project was administrated and supervised by KOD, HE, AH, MD, and BF. Resources and funding were acquired by MD and BF. KOD, HE, AH, MFS, MD, AR, and BF contributed to reviewing and editing the manuscript.

*Competing interests.* The corresponding author declares that none of the authors has any competing interests.

*Acknowledgements.* We want to thank the late Pär Jansson for initial discussions on combining M2PG1 with OpenDrift. KOD wants to thank Martin Arntsen for having the initial idea of using OpenDrift for this project and helping to sketch out the overarching themes of the framework. We want to thank Jon Albretsen and the Institute of Marine Research for sharing NorKyst simulation data. Finally, we would like to thank Tor Nordam and the two anonymous reviewers for their valuable comments that improved our paper. The language models GPT 3.0 and Claude Sonnet (v 3.0 and 3.5) assisted the python coding required to produce this work via the GitHub Co-pilot plugin in VSCode. GPT 3.0, and 4.0 was used to reformulate certain sentences and/or paragraphs in the text. This study was funded by the Research Council of Norway through EMAN7 (Environmental impact of Methane seepage and sub-seabed characterization at LoVe-Node 7, project number 320100) and ReGAME (Reliable global methane emissions estimates in a changing world, project number 325610).

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

| Location | Temp (°C) | $\phi$ (nM) | $k_{ox}$ $10^{-6} s^{-1}$ | $t_{0.5}$ (days) | Reference |
|---|---|---|---|---|---|
| **Oxic/anoxic interface** | | | | | |
| Cariaco Trench, Caribbean Sea | - | <12100 | 0.03 | 277 | Ward et al. (1987) |
| Saanich Inlet, British Columbia | 9 | <1580 | 0.02 | 535 | Ward et al. (1989) |
| Eastern Tropical North Pacific | - | 19 | 0.10 | 77 | Pack et al. (2015) |
| **Hydrothermal plume** | | | | | |
| Juan De Fuca v. | - | <390 | 1.74 | 5 | de Angelis et al. (1993) |
| **Man-made accidents** | | | | | |
| Deepwater Horizon, Gulf of Mexico | - | <183000 | 0.73 | 11 | Valentine et al. (2010) |
| North Sea gas blowout | 10 | <42097 | 0.41 | 20 | Steinle et al. (2016) |
| **Seep environment** | | | | | |
| Cape Lookout Bight, North Carolina | 23-27 | <740 | 0.08 | 107 | Sansone & Martens (1978) |
| Santa Barbara Channel, California | 5-16 | <1900 | 0.09 | 93 | Mau et al. (2012) |
| Boknis Eck, Baltic Sea | 1-3 | 300-466 | 0.50 | 16 | Steinle et al. (2017) |
| South China Sea | 2-5 | <1000 | 0.04 | 229 | Mau et al. (2020) |
| Hudson Canyon, US Atlantic | - | <335 | 0.93 | 9 | Weinstein et al. (2016) |
| Elson Lagoon, Alaska | -1.8 | <53.8 | 0.12 | 69 | Uhlig et al. (2018) |
| **Cold seeps - Svalbard Continental margin** | | | | | |
| Norskebanken | 4.7 | <83.1 | 0.06 | 125 | Sert et al. (2023) |
| Hinlopen Trough | 3.5 | <874 | 0.23 | 35 | De Groot et al. (2024) |
| Prins Karl Forland (2015) | ~3 | <334 | 0.98 | 8 | Gründger et al., (2021) |
| Prins Karl Forland (2016) | ~1.5 | <437 | 0.02 | 433 | Gründger et al., (2021) |
| Prins Karl Forland (2017) | ~3 | <262 | 0.02 | 385 | Gründger et al., (2021) |
| Prins Karl Forland | 1.6-4.8 | <524 | 0.21 | 38 | Gentz et al. (2014) |
| Hornsundbanken | >3 | <878 | 0.41 | 20 | Mau et al. (2017) |
| Isfjordenbanken | >3 | <100 | 0.62 | 13 | Mau et al. (2017) |
| Storfjordrenna | -0.5 | <82 | 0.22 | 36 | Sert et al. (2020) |
| Storfjorden | -1.5 | <72.3 | 0.35 | 23 | Mau et al. (2013) |

**Table E1.** Methane Oxidation Rate Coefficients ($k_{ox}$) in units of $10^{-6}$ s$^{-1}$ ($\mu$Hz) from various studies. We have obtained the maximum $k_{ox}$ reported in the studies unless ranges are given. Half-lives are calculated by solving for $\phi(t_{0.5}) = 0.5\phi_0$ in Eq. (E1), i.e. $t_{0.5} = \ln(2)/k_{ox}$. In Gründker et al., 2021 only May data was included from 2016 and the difference in turnover time between 2016 and 2017 is because the maximum rate coefficient was $1.85 \times 10^{-8}$ s$^{-1}$ in 2016 and $2.01 \times 10^{-8}$ s$^{-1}$ in 2017, but this difference is rounded off in the table.