# Peer review of "Modeling water column gas transformation, migration and atmospheric flux from seafloor seepage"

_EGUsphere, 2025_

## Author Response (AR1)

**Author's response to the reviews**

The following author's response will strongly rely on the replies to the individual author comment in the preprint discussion. Therefore, a large portion of this document will be copy-pasted from the discussion documents ('Reply on RC1', 'Reply on RC2', and 'Reply on RC5'). There are, however, some significant changes in how we have addressed certain comments since we discovered errors in our initial suggested modifications. In these cases, where the response is slightly modified from the discussion, we have highlighted the Review comment and Response headlines in red such that they will be easy for the reviewers to find. We apologize to the reviewers for this inconvenience. We have also added citations from the revised manuscript at certain places where it is unclear where and what edits were made. The document will be structured as follows:

1. Response to Reviewer 1.

2. Response to Reviewer 2.

3. Response to Reviewer 3.

4. Other changes

Finally, we would again like to thank all the reviewers for taking the time to review this manuscript. A considerable amount of work has been done and we do not take this for granted. Several comments were also vital to maintain the scientific quality of the work presented. Thank you.

**Response to Reviewer 1**

Dear Reviewer 1. We are very glad to hear you liked it. Thank you.

**Response to Reviewer 2**

We thank Reviewer 2 for the thoughtful and detailed comments on our manuscript. Below, we address the two general comments first, followed by the specific line-by-line comments.
* * *
**Reviewers general comment 1:**
*My main issues with the paper are first that I think the choice of rate coefficient for microbial oxidation could be discussed further. According to appendix E, half-lives found in the literature vary from 5 days to two years, and then to just consider a single rate coefficient which corresponds to a half-life of around 30 days seems a bit simple. I'm not saying you necessarily need to do more simulations, as I think the point of the paper is more to demonstrate a method than to give definite numbers, but I think this could be discussed further.*

**Response:**
We agree with the reviewer on this point and have expanded the discussion in the "Interpretation of the results" section, where we outline the importance and challenges of including microbial oxidation in the modelling framework. In addition, we conducted a set of sensitivity simulations with a range of rate coefficients to examine the impact on the fate of dissolved methane. This allows us to describe, at least in a semi-quantitative manner, the sensitivity of the modelling framework to different microbial oxidation rates.
* * *
**Reviewers general comment 2:**
*Second, the paper is quite long, and dedicates a lot of time to describing a partially new method for calculating concentrations. I am not completely convinced by the authors' arguments that this was necessary, and even if it was (necessary), I think much of Section 2.3 could go in the appendix, as it distracts from the main point. (In my opinion, the main point of the paper is that it's a good idea to combine a bubble model with a 3D transport-fate model for dissolved methane. The rest is just details.)*

**Response:**
We respectfully disagree, and we consider Section 2.3 to be one of the main contributions of the manuscript. This section addresses how realistic concentration estimates can be obtained using a Lagrangian particle dispersion model (LPDM). The choice of density estimator is very important in this context because the number of simulated particle trajectories is constrained by computational resources. As is well established, the commonly used histogram estimator performs poorly when particle numbers are limited, while kernel density estimators (KDEs) can provide substantial

improvements in both accuracy and computational efficiency (see e.g., de Haan, 1999; Vitali et al., 2006; Björnham et al., 2015; Sole-Mari et al., 2019; Egelrud, 2021; Barbero et al., 2023; Yang et al., 2025).

Key challenges with KDEs is the adaptation of bandwidths and boundary control. Both are particularly challenging in coastal ocean regions such as Northwestern Norway, where currents are typically highly dynamic and boundaries complex with numerous islands and fjords (the latter is also typically not an issue in atmospheric applications, where most of the currently available research are conducted). To our knowledge, no existing, currently freely available density estimator simultaneously (i) adapts bandwidths locally to ensure proper smoothing and (ii) handles coastal boundaries appropriately for these types of applications. We therefore concluded that developing a suitable kernel density estimator was necessary to realistically model concentrations in our setting, and that this methodological contribution would be valuable to the scientific community.

We consider Section 2.3 important because:
i) it represents a methodological advancement for concentration estimates relying on LPDMs in the ocean;
ii) it highlights limitations of current practices and proposes a solution;
iii) it is crucial for producing realistic concentration estimates while maintaining reasonable computation times, especially for long-duration simulations or computationally constrained scenarios.

We have revised the text and added further references to clarify this rationale (see the tracked-changes manuscript). In particular in section 2.3 where we now state

"Due to the extensive model domain and the need to obtain one estimate for every depth layer and time-step ($K \times N$ estimates), our density estimator needs to be fast and allow reliable density estimates from limited particle counts. It must also handle regions with low and high concentrations and concentration gradients as well as complex boundaries like fjords and islands. A commonly used density estimator in similar contexts is the histogram estimator, which unfortunately has several well-known limitations in these applications (the histogram estimator and its drawbacks are detailed in Appendix \ref{app:histogram_estimator}). Previous studies on concentration estimation from particle dispersion model data have shown that Kernel Density Estimators (KDEs) can offer far superior information exploitation than the histogram estimator and overcome many of its drawbacks \cite[see e.g][]{DeHaan1999,Vitali2006,Bjornham2015,Barbero2024,Yang2025}. One remaining challenge in our specific application, however, is the lack of available KDEs tailored to coastal ocean regions that appropriately adapt to spatial density variability (adaptive bandwidth) and complex boundary geometries (bathymetry). We therefore formulated a new 2-dimensional adaptive-bandwidth KDE to provide our density estimates."
* * *
**Reviewer comment (Line 7):**

*In what sense was the modelling "successful"? Did you compare against measurements or other estimate, or do you just mean that the model(s) ran successfully and produced reasonable output?*

**Response:**

We removed the word "successful."
* * *
**Reviewer comment (Line 22):**

*"Atmospheric measurements are currently the only approach..." Surely the current study is based on the assumption that also modelling is a possible approach to estimating atmospheric emissions from seep areas?*

**Response:**

We revised the text here and it now reads:
*"Estimation of total atmospheric gas emissions from seep areas \citep[e.g.][]{Myhre2016} rely largely on either ship measurements or large-scale atmospheric inversion models. The former of these approaches only give information on the local flux and require some sort of up-scaling, while the latter is unable to estimate dispersed sources and/or weaker point sources precisely due to its rough scale and inability to completely decouple atmospheric sources from sinks \citep{Thompson2014}."*

We also added references relating to these claims further into the introduction, see reply to reviewer 3 or revised manuscript.
* * *
**Reviewer comment (Line 33):**
*Alien gases?*

**Response:**
The text is now revised to:
*"Gas content in bubbles are constantly changing due to dissolution (gases in the bubble dissolve into the liquid) and exsolution (gases already dissolved in the liquid enter the bubble) driven by partial pressure gradients across the bubble rim."*
* * *
**Reviewer comment (Line 41):**
*What is "in situ data"?*

**Response:**

We revised the first two sentences of the paragraph to:

*"Our approach integrates a gas phase model with a hydrodynamic model using particle dispersion modelling to enable estimation of the three-dimensional (3D) distribution of gas in the water column and the total (free and diffusive) atmospheric gas release resulting from observed seabed seepage."*
* * *
*Personally, I find the notation with the square brackets a bit odd, though I suppose it is a matter of taste. And why use subscripts for particle number and square brackets for time? How about subscripts for particle and superscript for time? Or two subscripts? (There's lots of nice inspiration to be had from the world of general relativity: https://en.wikipedia.org/wiki/Christoffel_symbols#Definition_in_Euclidean_space )*

**Response:**

We considered several notational conventions. The use of square versus round brackets to distinguish between discrete and continuous arguments follows standard conventions in digital signal processing (e.g., Haykin & Van Veen, 1999). This approach avoids a proliferation of sub- and superscripts and helps readers distinguish between quantities defined in the discrete (gridded) domain denoted by square brackets, versus continuous (real-world, non-observable) domains denoted by round brackets. Subscripts are used for physical variables (e.g., particle indices, velocities), while brackets indicate position or distance. This distinction was also implemented to contribute to readability. A minor challenge arises in eqs. (11) – (15) though, where we have both a distance and positions, and in eq. (26) we perform a summation over both subscripts and brackets. However, we don't regard these issues as detrimental to the readability, as these are standard notation in signal processing and applied physics.

The suggested use of combined sub- and superscripts could possibly work in our case, but it would become cumbersome when combined with powers of variables (e.g., Eq. (19)). We therefore chose to maintain our present notation.

We included a short explanation of our regime when first encountered in the text, in line 118-119:

"Note that we throughout this manuscript will use square "$[\cdot]$" versus round "$(\cdot)$" brackets to distinguish between discrete and continuous spatiotemporal arguments, respectively."
* * *
**Reviewer comment (Lines 95–100):**

*Strictly speaking, this is not a correct description of the equation of motion for particles in OpenDrift. It solves an SDE, not an ODE, and there is no such thing as the diffusive velocity vector in the scheme used in OpenDrift. (If you try to calculate the diffusive velocity, you will find that it goes to infinity as dt ->0). The correct equation can be found in numerous papers, see for example Eq. (2) in Spivakovskaya et al. (2007): https://link.springer.com/article/10.1007/s10236-007-0102-9*

**Response:**

Thank you for pointing this out. In the reply in the discussion we had not captured the full depth of this comment. Therefore, the outlined change in the discussion reply is not how we have done it in the finalized manuscript. We instead modified the whole section and included the SDE in the manuscript. The paragraph now reads:

*"OpenDrift then calculates the trajectory of each particle individually by numerically solving a stochastic differential equation which is consistent with the Lagrangian representation of the advection-diffusion equation (see e.g. \cite{Spivakovskaya2007}). The drift in particle position $\boldsymbol{\eta}$ can be expressed as*

*%*

*\begin{equation}\label{eq:adv_diff}*

*d \boldsymbol{\eta} = \boldsymbol{U}\_\mu (\boldsymbol{\eta},t)dt + \boldsymbol{B}(\boldsymbol{\eta},t)d\boldsymbol{W}(t)*

*\end{equation}*

*%*

*where $\mathbf{U}\_\mu(\boldsymbol{\eta},t)$ represents displacement produced by the underlying (mean) velocity field and the second term represents displacement from random, diffusive processes and is composed of a diffusivity matrix $\boldsymbol{B}(\boldsymbol{\eta},t)$ and increments of a Wiener process $d\boldsymbol{W}(t)$. The advective term ($\mathbf{U}\_\mu\left(\boldsymbol{\eta},t\right)$) is determined by velocity fields obtained from the hydrodynamic model. OpenDrift represents the diffusivity $\boldsymbol{B}(\boldsymbol{\eta},t)$ as a diagonal matrix with a horizontal and a vertical diffusivity. If available, these diffusivities can be directly read from the hydrodynamic model output. Otherwise, OpenDrift can also estimate the diffusivity coefficients using one of several built-in parametrizations."*
* * *
**Reviewer comment (Lines 107–115):**

*I think these expressions are not quite consistent. On line 107 you say that the mass of particle \zeta, at time n, \Gamma_\zeta[n], is given by the previous mass and some functions. In Eq. (3), you say that \Gamma_\zeta[n] is defined by the release. At this stage, I am also confused about how you represent different release rates for different sites, and how you deal with the vertical variation in dissolution from the bubbles, if all particles are seeded with the same amount of mass, but presumably that will be described later?*

**Response:**

We agree that this was a bit unclear, and we have now restructured the relevant paragraphs for clarity. The text now reads:

*"The initial mass, i.e,. mass at release, of an arbitrary particle $\zeta$ is scaled such that the total released particle mass approximates the total number of moles of gas dissolved in the water column (from all modelled seeps combined) during a time interval $\Delta t$ centred on $t_n$. In practice, we distribute the integrated sum of modeled (using the gas phase model) injected gas molecules from $t_n-\Delta t/2$ to $t_n+\Delta t/2$ evenly over the seeded particles. Assuming that the seabed flux is stationary within the time interval $t_n\pm\Delta t/2$, the mass of a particle $\Gamma_\zeta$ seeded at time-step $n$ can be obtained by*

*%*

*\begin{equation}\label{eq:initial_mass}*

*$\Gamma_\zeta[n] = \frac{\Delta t\sum_{p=0}^{P}\Upsilon_p[n]}{S[n]}$*

*\end{equation}*

*%*

*where $\Upsilon_1[n], \Upsilon_2[n], \ldots, \Upsilon_{P}[n]$ [mol s$^{-1}$] are total injected dissolved gas from all $P$ modeled seeps. Approximation to the modelled dissolved gas release profiles at each modelled seep is achieved by seeding a different amount of particles at different depths."*
* * *
**Reviewer comment (Line 122):**

*Who refers to this as "particle death"? Very dramatic term, what's wrong with "particle removal"?*

**Response:**

We agree, and we have replaced "death" with "removal."
* * *
**Reviewer comment (Line 130):**

*Isn't there something wrong in Eq. (5)? Surely if \gamma_s is supposed to be based on the distance between particle s and particle \theta, then the position of particle s would be expected to appear on the right-hand side of the equation?*

**Response:**
Correct. The subscript $s$ was a remnant from a previous version. It has been corrected to \tau on the left-hand side.
* * *
**Reviewer comment (Line 148):**

*If I understand correctly, you said that you found histogram unsuitable, but you are essentially using a histogram in the vertical direction. Why not use 3D KDE?*

**Response:**
A full 3D KDE would indeed be desirable, but implementing a kernel estimator with adaptive bandwidth in 3D is considerably more complex and computationally demanding. For instance:

- we assumed isotropic kernels to simplify implementation and bandwidth estimation. In 3D, kernels would need to be anisotropic due to the very different scales in vertical vs horizontal dispersion. This would also reduce the robustness of the kernel bandwidth estimation, since we would be prohibited from averaging statistical properties (e.g., integral length scales) in all directions.

- boundary handling becomes more complex in 3D (top and bottom of water column in addition to lateral boundaries), increasing computational costs considerably if we keep using the kernel-specific boundary control algorithm presented in our paper.

Given these challenges, we considered our 2D layered approach sufficient for the present application. Moreover, horizontal dispersion completely dominates vertical dispersion, which reduces the noise introduced by using only a horizontal KDE. We acknowledge, however, that a 3D KDE would be a useful future development.
* * *
**Reviewer comment (Line 150):**

*Just a comment, but I think KDEpy is pretty efficient. Might be easier (and maybe faster?) than making a new implementation.*

**Response:**
We would like to refer also to our reply to General Comment 2 and the references cited therein.

In fact, we initially used KDEpy during the early stages of the project. While efficient and user-friendly (KDEfft in particular is very fast), KDEpy lacks several key features required for concentration estimates in coastal dispersion modelling:

1. **No local adaptation of kernel bandwidths.**
   Local bandwidth adaptation is crucial; without it, high-gradient regions become oversmoothed, while low-density regions become noisy. This issue is analogous to using fixed bin sizes in histograms. Adaptive bandwidths solve this problem by adjusting locally, whereas KDEpy does not.

2. **Insufficient boundary control.**
   In KDEpy, boundary correction is typically applied after the density estimate, e.g. by reflection. While acceptable in simple or infinite boundary problems, this fails in realistic coastal domains with islands, inlets, and fjords. Density can incorrectly "leak" over land masses into ocean areas on the other side of the landmass. Our method explicitly avoids this by enforcing boundary conditions at the kernel level, modifying each kernel shape according to the local geometry (Section 2.3.3).

3. **Inefficient variable bandwidth implementations.**
   KDEpy offers TreeKDE and NaiveKDE for varying bandwidths, however, varying bandwidth is not the same as adaptive variable bandwidth. Furthermore, these algorithms are significantly slower than our adaptive KDE implementation due to the grid projection. A simple performance test is available in our public repository: https://github.com/KnutOlaD/akd_estimator and a plot is provided further down in this report.
* * *
**Reviewer comment (Line 153):**
*Whether the density estimate is differentiable or not depends on the choice of kernel.*

**Response:**
We agree. The text has been revised to clarify this point.
* * *
**Reviewer comment (Line 157):**
*I don't think the variable V is defined.*

**Response:**
We thank the reviewer for noticing this. The variable $V$ has now been defined in the manuscript.
* * *
**Reviewer comment (Line 162):**

*I wouldn't worry about the kernel being consistent with diffusive transport. And in any case, if you wanted the kernel to be consistent with diffusion, wouldn't you have to let the bandwidth grow based on the diffusivity? And you also later truncate the kernel.*

**Response:**

We agree, and we have removed the sentence in question to avoid confusion.
* * *
**Reviewer comment (Lines 199, 202, and elsewhere):**

*Two different notations: N_{eff} and N_\mathrm{eff}.*

**Response:**

This inconsistency has been corrected. We now consistently use $N_\mathrm{eff}$.
* * *
**Reviewer comment (Line 260):**

*It's probably not important in the grand scheme of things, but redistributing the mass instead of mirroring fail in some simple test-cases. For example for uniform distribution on a bounded 1D domain, it will fail to reproduce a uniform distribution, but rather giving too low density near the boundaries.*

**Response:**

This is correct, and the kernel centre of mass will also be displaced more than ideal. However, because this redistribution is applied individually to every single kernel (approx. 300,000 times per time step in our test case) before summation - not just to the final (full) density estimate - the deviations become negligible in practice. See section 2.3.3.
* * *
**Reviewer comment (Line 276):**

*Wouldn't U_a more commonly be the wind at 10 m height?*

**Response:**

Yes. Both the equation and the ERA-Interim wind data correspond to 10 m above sea level. We have clarified this in the manuscript.
* * *
**Reviewer comment (Lines 292–296):**

*Do I understand correctly that any particle loses mass based on the sum of the flux \beta over the entire horizontal grid? So even a particle in an area with no wind will lose mass based on the average loss?*

**Response:**

We highly appreciate this observation, as it revealed an error in the text and the equation was also incorrect(!) In the model, particle weights are adjusted according to the flux from the local grid cell where the particle assuming no kernel distribution. The equation and text have been corrected accordingly and now reads:

"Loss of gas due to atmospheric flux is implemented by modifying the mass of all particles present in the surface layer. To ensure efficient computation and mass conservation, we assume that the entire contribution to the atmospheric flux from a surface layer particle occurs within the grid cell where that particle resides, disregarding the effects of mass distribution through the density kernels. Errors associated with this assumption are expected to be small, since wind and temperature fields and consequently, gas transfer velocities are generally smooth on typical kernel bandwidth scales. It is also mass conserving, because atmospheric flux varies linearly with dissolved gas concentration (Eq. (\ref{eq:atm_flux})). Furthermore, since grid cell concentration depends linearly on the total cell gas content (i.e., the sum of all particle masses in that cell) and the gridded gas transfer velocity, relative flux contributions from particles can be estimated using products of particle masses and cell specific gas transfer velocities. The mass loss due to atmospheric exchange for a surface layer particle $\alpha$ at time-step $n$ can then be expressed as

%

\begin{equation}\label{eq:atmospheric_loss}

\gamma_{\alpha}[n] =

\frac{\Gamma_\alpha[n]\widehat{\kappa}[c(\alpha),n]}

{\sum_{\zeta\in\mathcal{A}}\Gamma_\zeta[n]\widehat{\kappa}[c(\zeta),n]}

\sum_{i=1}^{I}\sum_{j=1}^{J}\widehat{\beta}[i,j,n],

\end{equation}

%

where $\alpha\in\mathcal{A}$ and $\mathcal{A}$ denotes the set of all surface-layer particles, and $c(\alpha)$ denotes the indices $i,j$ where particle $\alpha$ resides."
* * *
**Reviewer comment (Line 318):**
*Polynomial fit to visual observations of what?*

**Response:**
This refers to bubble size distributions. The manuscript has been revised here to clarify.
* * *
**Reviewer comment (Line 333 and Figure 6):**

*What type of average was used?*

**Response:**

A simple arithmetic mean of the tabulated values was used. This is now clarified in the text.
* * *
**Reviewer comment (Line 336):**

*Just out of curiosity: Does the deposited concentration actually decrease exponentially, or does it just look approximately exponential? Obviously it cannot be a _true_ exponential if it depends on environmental variables like temperature, but even for a theoretical case of a perfectly homogeneous water column, is it actually exponential?*

**Response:**

The solution to the transfer equation (Eq. (7) in Jansson et al., 2015), under the assumption of a single gas species and an otherwise stationary/constant environment would yield an exponential profile. However, when multiple gases, bubble rise velocity distributions, or variable environmental conditions are included, the result deviates from a strict exponential. In practice, an ambient undersaturated water column typically produces a dissolved profile that only exhibits exponential *appearance* from a simple visual inspection. We have revised the text to say "appears exponential" to reflect this more accurately.
* * *
**Reviewer comment (Line 343):**

*What are the units of the atmospheric fluxes?*

**Response:**

The units have been added in the manuscript.
* * *
**Reviewer comment (Line 348):**

*1,2,3,N, ? Missing some dots? But also, no need to repeat the concept of the indexing, but rather mention how many particles were used.*

**Response:**

We agree. The sequence has been shortened, and the redundant indexing explanation was removed.
* * *
**Reviewer comment (Lines 352–353):**

*No need to repeat terrain-following.*

**Response:**

The repetition has been removed.
* * *
**Reviewer comment (Line 357):**

*"therefore" ... strictly speaking it does not follow logically from the fact that the model is used by the authorities for acute happenings, that it is also the best model when running a modelling study several years later. (Note: I'm not saying that NorKyst isn't the best model, only that it doesn't follow logically from the statement)*

**Response:**

We agree. The sentence has been removed.
* * *
**Reviewer comment (Line 392):**

*Missing units on the rate coefficient.*

**Response:**

Units have been added in the revised manuscript
* * *
**Reviewer comment (Line 393):**

*How is the "mass modification term" used? And shouldn't it be negative?*

**Response:**

We revised and expanded this paragraph to better explain the implementation in the revised manuscript. The clarification should also now make it clear that the term is negative.
* * *
**Reviewer comment (Line 405):**

*Missing closing bracket. Also, I believe mol/L (molar) is a more common unit than mol/m3.*

**Response:**

We acknowledge that mol/L is common, but for consistency we chose to adhere strictly to SI units, expressed with powers ($mol\,m^{-3}$). This decision was motivated by the wide variety of unit conventions across the literature (e.g., seabed fluxes in ml/min, and diffusive fluxes with non-standard units as in Wanninkhof (2014) etc.)
* * *
**Reviewer comment (Lines 415–427):**

*I'm curious about the thickness of the vertical layers, and particularly the top layer, in the model used to calculate the flux from dissolved to the atmosphere. If this layer is to thin, the layer might be depleted on a faster timescale than mixing from the lower layers, since that mixing is modelled by particles that live in a different world, with a different timestep. On the other hand, if the layer is too thick, it will allow the escape of methane from too deep in the water column. In a hybdrid model like this, where the escape to the atmosphere and the transport of the particles are modelled by different approaches, this seems to me to be a point worth investigating and discussing.*

**Response:**

We agree that it is appropriate to highlight this issue. The text has been expanded to also discuss the influence of vertical layer thickness. The revised passage now reads:

*"Uncertainties in the eddy diffusivity, vertical transport and distribution are also expected to be large. The somewhat arbitrary choice of grid cell thickness can also modify the end result. If the grid cells are too thin, and the temporal resolution is too coarse, there is a risk of depletion of the surface layer between the model output time-steps. On the other hand, if the grid cells are too thick, one would incorporate $CH_4$ from depths where exchange with the atmosphere is unrealistic, thereby violating the assumptions of the atmospheric exchange bulk model. One can evaluate whether the surface thickness is sufficiently thick by comparing typical values for Eq. \ref{eq:atmospheric_loss} with the typical mass of surface layer particles and ensure that $\gamma_\alpha[n]<<\Gamma_\tau[n]$. "*
* * *
**Reviewer comment (Line 423):**
*Reads like the microbial flux is driven by wind speed.*

**Response:**
We agree that part of the sentence should not be there and has been removed
* * *
**Reviewer comment (Lines 424–425):**
*Do I read correctly that the loss due to particles leaving the domain is of the same order of magnitude as the loss to microbial oxidation? You have used a fairly high biodegradation rate, which is perfectly fair I think, but some studies claim to have found much slower oxidation rates (see for ekample https://pubs.acs.org/doi/full/10.1021/acs.est.7b02732). How would you use your model to study the case of slower biodegradation, without the loss due to particles leaving the domain being completely dominant?*

**Response:**

Yes, your interpretation is correct. This limitation arises because the model is constrained in both time and space by computational resources and the hydrodynamic domain boundaries. One option is to extrapolate by assuming that all methane molecules will eventually either reach the atmosphere or be consumed microbially, and that the relative partitioning observed within the model domain is representative of the long-term outcome. This assumption required for this approach means estimates will be increasingly less accurate for lower oxidation rates, since in those cases most methane leaves the domain unconsumed.

For exploring such scenarios, a 1D approach might provide a more robust framework for estimating the ultimate fate of methane. We have added a sentence to the conclusion to reflect this point and a reference to relevant work on this aspect (Nordam et al., 2015)
* * *
**Reviewer comment (Lines 456–457):**

*I would have thought that also the vertical eddy diffusivity used in the transport model is quite important. You say that the mass transfer coefficient is stated to have an uncertainty of 20%, my guess would be that the vertical eddy diffusivity from the ocean model, and the effect of that on the transport of methane towards the surface, has a (much) larger uncertainty than 20%. Any comments?*

**Response:**

We agree. Quantifying this uncertainty is not straightforward, as it depends on details of the hydrodynamic model and its coupling to the LPDM. Nevertheless, we have highlighted this issue in the discussion part with following sentence(s):

*"Uncertainties in the eddy diffusivity, vertical transport and distribution are also expected to be large. The somewhat arbitrary choice of grid cell thickness can also modify the end result (…)"*
* * *
**Reviewer comment (Lines 490–502):**

*I'm not sure I understand how the M2PG1 model works originally, or the implications of choosing a horizontal cell size. How can the original model take ambient concentrations into account for dissolution of methane from the bubbles, and for mass transfer of dissolved methane to the atmosphere, without calculating a concentration? And it also seems to me that the dissolution mostly happens near the sea floor, where the plume is narrow, while the mass transfer to the atmosphere only happens at the surface, where the plume is wide, so maybe the cell size should vary with depth? Finally, I wonder how sensitive the results are to this choice. Does much of the mass-transfer from the atmosphere happen in the M2PI1 model, or is that only calculated based on the*

*concentration fields from the particles? And to what degree does the ambient concentration in the water column hinder the dissolution of methane from the bubbles?*

**Response:**
Yes, ideally the cell size should vary with depth (which we have expressed in the appendix), but we use the estimated cell size at the mid-point of the water column, at 100 m depth. Furthermore, this is how M2PG1 works in combination with the concentration model:

1. M2PG1 calculates concentration and uses the background concentration as its initial condition. the concentration within the model domain is then modified iteratively by the dissolution processes. It is possible in M2PG1 to have dynamic background/boundary conditions where this can be modelled/measured, however, in the application/test presented in the manuscript, the boundary conditions are kept constant at the initial condition. An on-line coupling between the concentration/LPDM-based model and M2PG1 is desirable in future developments/applications.

2. The choice of model domain size becomes increasingly important when the concentrations are large, i.e., for more intense seeps than what we modelled here. The dissolution rates depend directly on the dissolved concentration in the water. The dissolved concentration depends on (among other things) the chosen domain size following Eq. (7) in Jansson et al., (2019) which motivated the addition of a grid cell size estimator.

3. The mass-transfer from the atmosphere happening in the M2PG1 model is not included since this is fully handled by the flux from the surface layer in the concentration model (the \beta[i,j,n] field).

We refer to Jansson et al., (2019) for further details.
* * *
**Reviewer comment (Line 533):**
*Should be "piece-wise constant function".*

**Response:**
We agree, this is corrected in the revised manuscript.
* * *
**Reviewer comment (Line 536):**
*I don't understand why the histogram estimator is "highly sensitive" to the position of the origin. I get that the results can look different if you shift the origin, but "highly sensitive"?*

**Response:**
The text has been revised to: *"can be sensitive."* This is also mainly a problem when the particle count is low.
* * *
**Reviewer comment (Lines 550 and 561):**

*What do the different numbers of particles (1900000 and 10^6) refer to?*

**Response:**

These numbers referred to the same case, and this was inconsistent in the submitted manuscript. We have corrected the manuscript and synthetic test case to consistently use and refer to 2000000 particles, which is both accurate and sounds less arbitrary.
* * *
**Reviewer comment (Line 551):**

*What does \mathcal{Z}^{100 \times 100} mean? To me it suggests some esoteric 10000 dimensional discrete space. Perhaps you mean i,j \in [1, 2, ... 100]?*

**Response:**

Yes, that was the intended meaning. The notation has been revised accordingly.
* * *
**Reviewer comment (Line 553):**

*Isn't there something odd in Eq. (C1)? The sum is over $t$, but there is no $t$ in the expression. Also, the units do not match. You mention a timestep later, but it should be included here to obtain displacement from velocity, and the random vector should be scaled by sqrt(dt) to match the units of sqrt(D).*

**Response:**

We agree. The equation has been corrected and clarified. The omission of time and normalization with $\sqrt{\Delta t}$ was an oversight caused by the unity timestep used in testing. The revised version should now be consistent and dimensionally correct.
* * *
**Reviewer comment (Lines 565–567):**

*Again, I would suggest including KDEpy (https://kdepy.readthedocs.io/en/latest/) in the comparison. I'm not sure how it will compare, but in my limited experience I've found it quite fast. Also, I'm not sure if Silverman's rule is a very relevant option, I believe it is well known to be less than optimal for multi-modal distributions.*

**Response:**

We acknowledge this suggestion. As noted earlier, we consider that KDEpy does not provide the capabilities (adaptive bandwidth estimation and boundary control) needed for our application. When it comes to performance, we have included simple performance testing in the github repository where it is possible to compare with both the KDEs included in Scipy as well as KDEpy. We now refer to this in the manuscript. Performance-wise the grid projected akd presented in our manuscript is slightly slower

than the FFT implementation in KDEpy, which omits bandwidth adaptation and does not allow variable kernel bandwidth (see also figure below for computation time of densities from a 2d gaussian distributed particle cloud).

[Figure]

When it comes to the application of Silverman's rule in our estimator: It is true that Silverman's rule is less than optimal for multi-modal distributions. However, we use Silverman's rule only locally within each adaptation window, for each grid cell centre wherein the assumption of a unimodal distribution is approximately valid (in essence we use Silverman's rule once for each particle containing grid cell to estimate the bandwidths from which we construct the concentration field – which can indeed be multi-modal!). We are now also explicit about this assumption and have included, when describing the adaptation window "adapting $h$ locally for a square shaped horizontal "adaptation" grid of size $P \times P$ surrounding each particle containing grid cell where we assume near normal, unimodal distribution"
* * *
**Reviewer comment (Line 573):**
*"theoretically ideal". Only a minor detail, but I'm not sure if this would be theoretically ideal (depends on the theory, I guess). In a particle model, where the particles are transported by the diffusivity, this increases the variance of the overall distribution. But the kernel bandwidth also increases the variance of the overall distribution. I seem to rememeber that it is easy to show (at least of all particles have the same kernel and bandwidth) that the variance of the distribution obtain by KDE is the sum of the variance of the particle positions, and the variance of the kernels. So I wonder if this "theoretically ideal" bandwidth wouldn't in fact double-count the diffusion? You could test it against an analytical solution of the diffusion equation in 1D.*

**Response:**
We removed "theoretically ideal", however, we are not sure we quite follow the argument about double counting and cannot find any published proof pointing in this direction. Nonetheless, we acknowledge that this is not the right place for delving into proofs of what's theoretically ideal or not in selecting the kernel bandwidths and have removed "theoretically ideal" from the manuscript.

We also refer to the discussions presented in Björnham, et al., (2015) who allows the bandwidth to grow with the diffusivity in their applied atmospheric model.
* * *
**Reviewer comment (Line 590):**
*Some words missing at the end here?*

**Response:**
Correct. The sentence has been corrected.
* * *
**Reviewer comment (Figure C2):**
*The histogram example uses very small cells (10000 cells and only 1000 particles), it would probably be closer to the "ground truth" with larger cells.*

**Response:**
Thank you for pointing this out. The caption has been corrected to clarify that the left-hand panel shows the histogram estimate of the test dataset. The "ground truth" was generated using 2,000,000 particles, which should yield a quite smooth distribution for 10000 cells.
* * *
**Reviewer comment (Line 634):**
*I suggest a manual proofread as well, there are still a few typos here and there (including on this line).*

**Response:**
A thorough manual proofread has been conducted, and typographical and grammatical errors have been corrected throughout and we have also modified several sentences to improve readability.
* * *
**Reviewer comment regarding the advantages of a 3D-model**

*Just one thing I forgot to mention in the review comments, that you could perhaps also discuss: What are the advantages of 3D modelling compared to 1D modelling? You mention some of the drawbacks (considerable computational effort, loss of mass*

*through particles exiting the domain), so it would be interesting to discuss if it is worth the effort.*

**Response:**

The main advantages of the 3D modelling is now addressed in the conclusion, being i) the ability to incorporate more advanced hydrodynamic processes via established 3d hydrodynamic models ii) the 3D concentration field is necessary in a wide range of other applications/studies of particular processes.
* * *
**Reviewer comment (Figure 9):**
*Oh, and one more small point: I think the units on the colorbar in Fig. 9 are wrong, the numbers seem to be in mol/L not mol/m3?*

**Response:**
Regarding the units in Figure 9 we believe that those are correct. We changed the figure caption to hopefully better explain the contents of the figure. We aim to display the anomaly from the background concentrations. The background concentration is $3 \cdot 10^{-6}$ moles/m$^{-3}$, i.e. 3 nmol/L.
* * *
**Response to Reviewer 3**

The authors thank the reviewer for a thorough evaluation of our manuscript. Below we provide a point-by-point response.
* * *
**Reviewer comment (Line 128):**
*"The redistribution is weighted according to the inverse distance from the dying particle within a user defined distance limit dmax,: How does the diffusive transport of particles that is adjusted by this re-distribution of mass and affect the model results?*

**Response:**
We agree that this should be discussed, and the text has been revised and now reads:
*"This solution changes the problem of non-physical loss of dissolved gas to one of non-physical redistribution. This can affect model results by shifting particle mass towards the seed location, since the density of particles are in general higher closer to the release point. However, we consider this artifact less problematic than mass simply disappearing."*
* * *
**Reviewer comment (Page 5, last line):**
*"This solution changes the problem of non-physical loss of dissolved gas to one of non-physical redistribution, which is generally considerably less problematic."*

*How do you say this is less problematic? Based on any evidence or proof?*

**Response:**
We appreciate the question. While it is difficult to "prove" this formally in general, our reasoning is as follows: if a particle with mass $M$ is completely removed, this results in a direct error of $M$ in the concentration field. By contrast, redistributing the mass preserves the total amount, though at the expense of potentially shifting its spatial distribution. We therefore expect the numerical error in the field to be smaller.

We acknowledge that in very specific, artificial cases - such as a constant, noise-free current where particles are repeatedly removed at the same point - redistribution could theoretically cause severe local artifacts. However, we consider this scenario very unlikely in the applications discussed in our paper.
* * *
**Reviewer comment (Lines 35–40):**
*"Our aim is to provide a framework which can integrate all key processes governing free and dissolved transport and transformation of seeped gas to provide a full 3-D concentration field in the water column and total atmospheric release estimates." There*

*have been previous modeling studies in literature focusing on these aspects. See the studies: https://pubs.acs.org/doi/full/10.1021/acs.estlett.3c00493, https://www.nature.com/articles/s41467-024-53780-7, https://sintef.brage.unit.no/sintef-xmlui/handle/11250/2730544 , https://pubs.acs.org/doi/full/10.1021/acs.est.5c03297*

**Response:**
We thank the reviewer for these valuable references. To better reflect previous work done on this topic we have revised the whole second paragraph of the introduction. It now reads:

*"Gas released at the seabed can enter the atmosphere directly as free gas (bubbles) or via diffusive equilibrium of dissolved gas that has reached the sea surface. To estimate the total atmospheric emissions from a seabed seep and its dissolved distribution in the water column, one must be able to model both pathways simultaneously. Gas content in bubbles is constantly changing due to dissolution (gases in the bubble dissolve in the liquid) and exsolution (gases already dissolved in the liquid enter the bubble) driven by partial pressure gradients across the bubble rim. Additionally, chemical and biological processes can modify local dissolved gas content.Estimating the gas distribution in the water column and total atmospheric flux therefore requires a flexible framework which can integrate processes governing the gas phase dynamics and the hydrodynamics, accommodate atmospheric exchange, and other phenomena that modify water column gas content. Previous modelling efforts have typically focused on single gas phase frameworks including only selected processes \citep[e.g][]{McGinnis2006,Graves2015,Silyakova2020}, however, key steps towards modeling the complete system have been made recently in \cite{Dissanyake2023} and \cite{Nordam2025}. We aim to further expand on these studies from a methodological perspective and provide a pilot framework which can integrate all key processes governing free and dissolved transport and transformation of seeped gas and give a realistic estimate of the time varying 3-dimensional (3D) water column concentration field and 2-dimensional (2D) atmospheric release field."*
*"Even though previous modelling efforts have mainly focused on modeling individual pathways \citep[e.g.][]{McGinnis2006, Graves2015, Silyakova2020}, key steps toward modelling the complete system have been made in \cite{Dissanyake2023} and \cite{Nordam2025}. We aim to expand on these studies from a methodological perspective, to provide a pilot framework that integrates all key processes governing free and dissolved transport and transformation of seeped gas, yielding a full 3-D concentration field in the water column and estimates of total atmospheric release."*

To further emphasize the contribution of Dissanyake et al. (2023), we added a reference to their study in the final paragraph in the introduction section (since the approach is similar).

Regarding the SINTEF report, we note that Nordam et al. (2025) builds extensively on this work and has undergone peer review. For this reason, we have chosen to cite the readily available paper by Nordam et al. (2025) rather than the SINTEF report.
* * *
**Reviewer comment (Line 345):**
*"Modeling step, were calculated using the dissolved gas profiles" Are these modeled or measured?*

**Response:**
They are modeled. The sentence has been revised for clarity:
*"Dissolved gas injection rates, which are needed as input in the particle dispersion modelling step, were calculated using modelled (by M2PG1) dissolved gas profiles (not shown) and Eq. (...)."*
* * *
**Reviewer comment (Line 505):**
*In the model presented, the vertical binning size is dependent on the bubble rising speeds wo. The bubble size which controls the rising speed changes with pressure changes and the mass transfer at different depths. Hence the bubble size distribution (BSD) changes at different depths. BSD at what depth level was used when you decided the vertical bin size and why?*

**Response:**
This is correct. This is true and the note about the assumed unchanged BSD should have been included. We included the following in the revised manuscript:
*"Note that the BSD is expected to change with height above the seafloor (which also changes $P[w_o]$). For the purpose of this calculation, however, we assume the BSD remains constant"*
* * *
**Reviewer comment (Line 550):**
*"OpenDrift by seeding N = 1900000 particles". This is a very large number of particles that were used in the simulations. How did you decide (what is the basis) the number of particles to be used? and how much computer resources needed for these simulations and time taken. It would be good to give an idea of this to the readers.*

**Response:**
The number just has to be sufficiently large to ensure that the "ground truth" histogram estimator manages to obtain a reasonable density estimate over the chosen grid (the non-normalized error is proportional to $1/N_{a,b}^{0.5}$ where $N_{a,b}$ is the number of particles in the cell). To make it sound less arbitrary we changed the particle number to

2000000 (400 timesteps and 5000 at each step) in the revised manuscript. However, it has only negligible effects on the end result.

We also added some performance metrics to the manuscript and referred to the public GitHub repository, where simple performance tests are available for both the synthetic PDM toy model and different KDEs.
* * *
**Reviewer comment (Line 620):**
*In this section where you describe the oxidation rates you are presenting several ranges. For example rates of CH4 oxidation*

*Varying from 10−8 to 10−2 nM s−1,) and ((kox) range from 0.02·10−6 to 1.74·10−6 s−1,. You should state which numbers were used in your study in the Norwegian waters and justify the reasons for choosing these numbers as there is a large variation.*

**Response:**
The values applied in our case study are stated in Section 3.3 and also later in Section 3. To improve clarity, we added the sentence:
*"For the application offshore northwestern Norway, we used the average of all compiled values."*

We acknowledge the lack of local measurements and therefore have no stronger justification beyond using an average. This is also discussed in our reply to General Comment 1 from Reviewer 2, which partly addresses this limitation.
* * *
**Reviewer comment (Section 4, Conclusions):**
*I expect large variability in the biodegradation rates will affect the results presented to a large extent. I believe should be discussed in the manuscript and included in the conclusion. Please see the recent studies presented in https://sintef.brage.unit.no/sintef-xmlui/handle/11250/2730544 and https://pubs.acs.org/doi/full/10.1021/acs.est.5c03297*

**Response:**
We agree. The "Interpretation of the results" section now includes an expanded discussion on the importance and challenges of incorporating microbial oxidation into the modelling framework.

We also performed a sweep of rate coefficients and examined the resulting sensitivity of atmospheric fluxes and concentrations. This analysis is now included in the revised manuscript, and in the conclusions we added the following:

*"In particular, mass loss due to microbial oxidation pose a significant challenge since rate coefficients are shown to exhibit large variability which cause considerable*

*differences in the modeled atmospheric fluxes and concentrations. Current parameterizations of mass loss due to atmospheric ventilation are also simple and developed for large scale ocean regions, not coastal areas."*

**Other changes/comments**

During the revision process, we discovered several smaller errors and inconveniences in the manuscript. We also added some clarifications, e.g. pointing out the physical rationale behind having a kernel specific boundary control in line 301 in the track-changed manuscript or clarification around how the mass modification terms are added in line 148-152 (track changed manuscript). Additionally, since we included a small sensitivity analysis, the abstract had to be revised to accommodate the new results.

**References:**

Barbero, D., Ribstein, B., Nibart, M., Carissimo, B., & Tarniewicz, J. (2024). Reduction of simulation times by application of a kernel method in a high-resolution Lagrangian particle dispersion model. Air Quality, Atmosphere & Health, 17, 2105-2117.

Björnham, O., Brännström, N., Grahn, H., Lindgren, P., & Von Schoenberg, P. (2015). Post-processing of results from a particle dispersion model by employing kernel density estimation.

De Haan, P. (1999). On the use of density kernels for concentration estimations within particle and puff dispersion models. Atmospheric Environment, 33(13), 2007-2021.

Egelrud, D. (2021). Kernel density estimators as a tool for atmospheric dispersion models (Publication No. 1592045) [Master dissertation, Umeå University]. Digitala Vetenskapliga Arkivet.

Haykin, S., and Van Veen, B. (1999). Signals and Systems (1st ed.), Wiley.

Jansson, P., Ferré, B., Silyakova, A., Dølven, K., & Omstedt, A. (2019). A new numerical model for understanding free and dissolved gas progression toward the atmosphere in aquatic methane seepage systems. Limnology and Oceanography: Methods, 17 (3).

Sole-Mari, G., Bolster, D., Fernández-Garcia, D., & Sanchez-Vila, X. (2019). Particle density estimation with grid-projected and boundary-corrected adaptive kernels. Advances in Water Resources, 131, 103382.

Vitali, L., Monforti, F., Bellasio, R., Bianconi, R., Sachero, V., Mosca, S., & Zanini, G. (2006). Validation of a Lagrangian dispersion model implementing different kernel methods for density reconstruction. Atmospheric Environment, 40(40), 8020-8033.

Yang, L., Fang, S., Wang, Z., Song, J., Li, X., & Chen, Y. (2025). Optimizing and evaluating multiple kernel density estimators for local-scale atmospheric dispersion modeling at a representative AP1000 nuclear power plant site in China. Nuclear Engineering and Technology, 103880.